# Simple, Scalable and Effective Clustering via One-Dimensional Projections

**Moses Charikar**
Stanford University
Stanford, CA
`moses@cs.stanford.edu`

**Monika Henzinger**
Institute of Science and Technology Austria (ISTA)
Klosterneuburg, Austria
`monika.henzinger@ist.ac.at`

**Lunjia Hu**
Stanford University
Stanford, CA
`lunjia@stanford.edu`

**Maximilian Vötsch**
Faculty of Computer Science
Doctoral School of Computer Science DoCS Vienna
University of Vienna
Vienna, Austria
`maximilian.voetsch@univie.ac.at`

**Erik Waingarten**
Department of Computer and Information Sciences
University of Pennsylvania
Philadelphia, PA
`ewaingar@seas.upenn.edu`

## Abstract

Clustering is a fundamental problem in unsupervised machine learning with many applications in data analysis. Popular clustering algorithms such as Lloyd's algorithm and $k$-means++ can take $\Omega(ndk)$ time when clustering $n$ points in a $d$-dimensional space (represented by an $n \times d$ matrix $X$) into $k$ clusters. In applications with moderate to large $k$, the multiplicative $k$ factor can become very expensive. We introduce a simple randomized clustering algorithm that provably runs in expected time $O(\mathsf{nnz}(X) + n \log n)$ for arbitrary $k$. Here $\mathsf{nnz}(X)$ is the total number of non-zero entries in the input dataset $X$, which is upper bounded by $nd$ and can be significantly smaller for sparse datasets. We prove that our algorithm achieves approximation ratio $\widetilde{O}(k^4)$ on any input dataset for the $k$-means objective. We also believe that our theoretical analysis is of independent interest, as we show that the approximation ratio of a $k$-means algorithm is approximately preserved under a class of projections and that $k$-means++ seeding can be implemented in expected $O(n \log n)$ time in one dimension. Finally, we show experimentally that our clustering algorithm gives a new tradeoff between running time and cluster quality compared to previous state-of-the-art methods for these tasks.

## 1 Introduction

Clustering is an essential and powerful tool for data analysis with broad applications in computer vision and computational biology, and it is one of the fundamental problems in unsupervised machine learning. In large-scale applications, datasets often contain billions of high-dimensional points. Grouping similar data points into clusters is crucial for understanding and organizing datasets. Because of its practical importance, the problem of designing efficient and effective clustering algorithms has attracted the attention of numerous researchers for many decades.

37th Conference on Neural Information Processing Systems (NeurIPS 2023).

One of the most popular algorithms for the $k$-means clustering problem is Lloyd's algorithm [52], which seeks to locate $k$ centers in the space that minimize the sum of squared distances from the points of the dataset to their closest center (we call this the "$k$-means cost"). While finding the centers minimizing the objective is NP-hard [3], in practice we can find high-quality sets of centers using Lloyd's iterative algorithm. Lloyd's algorithm maintains a set of $k$ centers. It iteratively updates them by assigning points to one of $k$ clusters (according to their closest center), then redefining the center as the points' center of mass. It needs a good initial set of centers to obtain a high-quality clustering and fast convergence. In practice, the $k$-means++ algorithm [6], a randomized *seeding* procedure, is used to choose the initial $k$ centers. $k$-means++ achieves an $O(\log k)$-approximation ratio in expectation, upon which each iteration of Lloyd's algorithm improves.[1] Beyond their effectiveness, these algorithms are simple to describe and implement, contributing to their popularity.

The downside of these algorithms is that they do not scale well to massive datasets. A standard implementation of an iteration of Lloyd's algorithm needs to calculate the distance to each center for each point in the dataset, leading to a $\Theta(ndk)$ running time. Similarly, the standard implementation of the $k$-means++ seeding procedure produces $k$ samples from the so-called $D^2$ distribution (see Appendix A for details). Maintaining the distribution requires making a pass over the entire dataset after choosing each sample. Generating $k$ centers leads to a $\Theta(ndk)$ running time. Even for moderate values of $k$, making $k$ passes over the entire dataset can be prohibitively expensive.

One particularly relevant application of large-scale $k$-means clustering is in approximate nearest neighbor search [66] (for example, in product quantization [44] and building inverted file indices [16]). There, $k$-means clustering is used to compress entire datasets by mapping vectors to their nearest centers, leading to billion-scale clustering problems with large $k$ (on the order of hundreds or thousands). Other applications on large datasets requiring a large number of centers may be spam filtering [61, 65], near-duplicate detection [42], and compression or reconciliation tasks [62]. New algorithmic ideas are needed for these massive scales, and this motivates the following challenge:

> Can we design a simple, practical algorithm for $k$-means that runs in time roughly $O(nd)$, independent of $k$, and produces high-quality clusters?

Given its importance in theory and practice, a significant amount of effort has been devoted to algorithms for fast $k$-means clustering. We summarize a few of the approaches below with the pros and cons of each so that we may highlight our work's position within the literature:

A. **Standard $k$-means++**: This is our standard benchmark. **Plus:** Guaranteed to be an $O(\log k)$-approximation [6]; outputs centers, as well as the assignments of dataset points to centers. **Minus:** The running time is $O(ndk)$, which is prohibitively expensive in large-scale applications.

B. **Using Approximate Nearest Neighbor Search**: One may implement $k$-means++ faster using techniques from approximate nearest neighbor search (instead of a brute force search each iteration). **Plus:** The algorithms with provable guarantees, like [26], obtain an $O_\varepsilon(\log k)$-approximation. **Minus:** The running time is $\widetilde{O}(nd + (n \log(\Delta))^{1+\varepsilon})$, depending on a dataset dependent parameter $\Delta$, the ratio between the maximum and minimum distances between input points. The techniques are algorithmically sophisticated and incur extra poly-logarithmic factors (hidden in $\widetilde{O}(\cdot)$), making the implementation significantly more complicated.

C. **Approximating the $D^2$-Distribution**: Algorithms that speed up the seeding procedure for Lloyd's algorithm or generate fast coresets (we expand on this below) have been proposed in [8, 7, 10]. **Plus:** These algorithms are fast, making only one pass over the dataset in time $O(nd)$. (For [8, 7], there is an additional additive $O(k^2d)$ term in the running time). **Minus:** The approximation guarantees are qualitatively weaker than the approximation of $k$-means clustering. They incur an additional additive approximation error that grows with the entire dataset's variance (which can lead to an arbitrarily large error; see Section 3). These algorithms output a set of $k$ centers but not the cluster assignments. Naively producing the assignments would take time $O(ndk)$.[2]

---

[1]Approximation is with respect to the $k$-means cost. A $c$-approximation has $k$-means cost, which is at most $c$ times larger than the optimal $k$-means cost.

[2]One may use approximate nearest neighbor search techniques to improve on the $O(ndk)$ running time. However, as discussed above, approximate nearest neighbor search adds a significant layer of complexity (and approximation).

**Coresets.** At a high level, coresets are a dataset-reduction mechanism. A large dataset $X$ of $n$ points in $\mathbb{R}^d$ is distilled into a significantly smaller (weighted) dataset $Y$ of $m$ points in $\mathbb{R}^d$, called a "coreset" which serves as a good proxy for $X$, i.e., the clustering cost of any $k$ centers on $Y$ is approximately the cost of the same centers on $X$. We point the reader to [9, 35] for a recent survey on coresets. Importantly, coreset constructions (with provable multiplicative-approximation guarantees) require an initial approximate clustering of the original dataset $X$. Therefore, any fast algorithm for $k$-means clustering automatically speeds up any algorithmic pipeline that uses coresets for clustering — looking forward, we will show how our algorithm can significantly speed up coreset constructions without sacrificing approximation.

Beyond those mentioned above, many works seek to speed up $k$-means++ or Lloyd iterations by maintaining some nearest neighbor search data structures [58, 54, 46, 45, 33, 60, 40, 59, 71, 30, 29, 13, 55, 28, 18], or by running some first-order methods [64]. These techniques do not give provable guarantees on the quality of the $k$-means clustering or on the running time of their algorithms.

**Theoretical Results.** We give a simple randomized clustering algorithm with provable guarantees on its running time and approximation ratio without making any assumptions about the data. It has the benefit of being fast (like the algorithms in Category C above) while achieving a multiplicative error guarantee without additional additive error (like the algorithms in Category B above).

- The algorithm runs in time $O(nd + n \log n)$ irrespective of $k$. It passes over the dataset once to perform data reduction, which gives the $nd$ factor plus an additive $O(n \log n)$ term to solve $k$-means on the reduced data, producing $k$ centers and cluster assignments. On sparse input datasets, the $nd$ term becomes $\mathsf{nnz}(X)$, where $\mathsf{nnz}(X)$ is the number of non-zero entries in the dataset. Thus, our algorithm runs in $O(\mathsf{nnz}(X) + n \log n)$ time on sparse matrices.

- The algorithm is as simple as the $k$-means++ algorithm while significantly more efficient. The approximation ratio we prove is $\mathrm{poly}(k)$, which is worse than the $O(\log k)$-approximation achieved by $k$-means++ but it is purely multiplicative (see the remark below on improving this to $O(\log k)$). It does not incur the additional additive errors from the fast algorithms in [8, 7, 10].

Our algorithm projects the input points to a random one-dimensional space and runs an efficient $k$-means++ seeding after the projection. For the approximation guarantee, we analyze how the approximation ratio achieved after the projection can be transferred to the original points (Lemma 2.5). We bound the running time of our algorithm by efficiently implementing the $k$-means++ seeding in one dimension and analyzing the running time via a potential function argument (Lemma 2.4). Our algorithm applies beyond $k$-means to other clustering objectives that sum up the $z$-th power of the distances for general $z \geq 1$, and our guarantees on its running time and approximation ratio extend smoothly to these settings.

**Improving the Approximation from** $\mathrm{poly}(k)$ **to** $O(\log k)$**.** The approximation ratio of $\mathrm{poly}(k)$ may seem significantly worse than the $O(\log k)$ approximations achievable with $k$-means++. However, we can improve this to $O(\log k)$ with an additional, additive $O(\mathrm{poly}(kd) \cdot \log n)$ term in the running time. Using previous results discussed in Appendix A.2 (specifically Theorem A.2), a multiplicative $\mathrm{poly}(k)$-approximation suffices to construct a coreset of size $\mathrm{poly}(kd)$ and run $k$-means++ on the coreset. Constructing the coreset is simple and takes time $\mathrm{poly}(kd) \cdot \log n$ (by sampling from an appropriate distribution); running $k$-means++ on the coreset takes $\mathrm{poly}(kd)$ time (with no dependence on $n$). Combining our algorithm with coresets, we get a $O(\log k)$-approximation in $O(\mathsf{nnz}(X)) + O(n \log n) + \mathrm{poly}(kd) \cdot \log n$ time. Notably, these guarantees cannot be achieved with the additive approximations of [8, 7, 10].

**Experimental Results.** We implemented our algorithm, as well as the lightweight coreset of [10] and $k$-means++ with sensitivity sampling [15]. We ran two types of experiments, highlighting various aspects of our algorithm. We provide our code in the supplementary material. The two types of experiments are:

- **Coreset Construction Comparison**: First, we evaluate the performance of our clustering algorithm when we use it to construct coresets. We compare the performance of our algorithm to $k$-means++ with sensitivity sampling [9] and lightweight coresets [10]. In real-world, high-dimensional data, the cost of the resulting clusters from the three algorithms is roughly the same. However, ours and the lightweight coresets can be significantly faster (ours is up to **190x** faster than $k$-means++,

see Figure 2 and Table 1). The lightweight coresets can be faster than our algorithm (between 3-5x); however, our algorithm is "robust" (achieving multiplicative approximation guarantees).[3] Additionally, we show that the clustering from lightweight coresets can have an arbitrarily high cost for a synthetic dataset. On the other hand, our algorithm achieves provable (multiplicative) approximation guarantees irrespective of the dataset (this is demonstrated in the right-most column of Figure 2).

- **Direct k-means++ comparison**: Second, we compare the speed and cost of our algorithm to k-means++[6] as a stand-alone clustering algorithm (we also compare two other natural variants of our algorithm). Our algorithm can be up to **800x** faster than $k$-means++ for $k = 5000$ and our slowest variant up to **100x** faster (Table 1). The cost of the cluster assignments can be significantly worse than that of $k$-means++ (see Figure 3). Such a result is expected since our theoretical results show a $\text{poly}(k)$-approximation. The other (similarly) fast algorithms (based on approximating the $D^2$-distribution) which run in time $O(nd)$ [8, 7] do not produce the cluster assignments (they only output $k$ centers). These algorithms would take $O(ndk)$ time to find the cluster assignments — this is precisely the computational cost our algorithm avoids.

We do not compare our algorithm with [26] nor implement approximate nearest neighbor search to speed up $k$-means++ for the following reasons. The algorithm in [26] is significantly more complicated, and there is no publicly available implementation. In addition, both [26] and approximate nearest neighbor search incur additional poly-logarithmic (or even $n^{o(1)}$-factors for nearest neighbor search over $\ell_2$ [4]) which add significant layers of complexity to the implementation and make a thorough evaluation of the algorithm significantly more complicated. Instead, our current implementation demonstrates that a simple, one-dimensional projection and $k$-means++ on the line enables dramatic speedups to coreset constructions without sacrificing approximation quality.

**Related Work.** Efficient algorithms for clustering problems with provable approximation guarantees have been studied extensively, with a few approaches in the literature. There are polynomial-time (constant) approximation algorithms (an exponential dependence on $k$ is not allowed) (see [50, 17, 2, 39] for some of the most recent and strongest results), nearly linear time $(1 \pm \varepsilon)$-approximations with running time exponential in $k$ which proceed via coresets (see [41, 19, 36, 37, 15, 9, 27, 25] and references therein, as well as the surveys [1, 35]), and nearly-linear time $(1 \pm \varepsilon)$-approximations in fixed / low-dimensional spaces [5, 48, 68, 38, 24, 22, 23]. Our $O(n \log n)$-expected-time implementation of $k$-means++ seeding achieves an $O(\log k)$ expected approximation ratio for $k$-median and $k$-means in one dimension. We are unaware of previous work on clustering algorithms running in time $O(n \log n)$.

Another line of research has been on dimensionality reduction techniques for $k$-means clustering. Dimensionality reduction can be achieved via PCA based methods [31, 37, 21, 67], or random projection [21, 11, 53]. For random projection methods, it has been shown that the $k$-means objective is preserved up to small multiplicative factors when projecting onto $O_\varepsilon(\log(k))$ dimensional space. Additional work has shown that dimensionality reduction can be performed in $O(\text{nnz}(A))$ time [51]. To the best of our knowledge, we are the first to show that clustering objectives such as $k$-median and $k$-means are preserved up to a $\text{poly}(k)$ factor by one-dimensional projections.

Some works show that the $O(\log k)$ expected approximation ratio for $k$-means++ can be improved by adding local search steps after the seeding procedure [49, 20]. In particular, Choo et al. [20] showed that adding $\varepsilon k$ local search steps achieves an $O(1/\varepsilon^3)$ approximation ratio with high probability.

Several other algorithmic approaches exist for fast clustering of points in metric spaces. These include density-based methods like DBSCAN [34] and DBSCAN++ [43] and the line of heuristics based on the Partitioning Around Medoids (PAM) approach, such as FastPAM [63], Clarans [56], and BanditPAM [69]. While these algorithms can produce high-quality clustering, their running time is at least linear in the number of clusters (DBSCAN++ and BanditPAM) or superlinear in the number of points (DBSCAN, FastPAM, Clarans).

---

[3]Recall that the lightweight coresets incur an additional additive error which can be arbitrarily large.

## 2 Overview of Our Algorithm and Proof Techniques

Our algorithm, which we call PRONE (PRojected ONE-dimensional clustering), takes a random projection onto a one-dimensional space, sorts the projected (scalar) numbers, and runs the $k$-means++ seeding strategy on the projected numbers. By virtue of its simplicity, the algorithm is scalable and effective at clustering massive datasets. More formally, PRONE receives as input a dataset of $n$ points in $\mathbb{R}^d$, a parameter $k \in \mathbb{N}$ (the number of desired clusters), and proceeds as follows:

1. Sample a random vector $v \in \mathbb{R}^d$ from the standard Gaussian distribution and project the data points to one dimension along the direction of $v$. That is, we compute $x'_i = \langle x_i, v \rangle \in \mathbb{R}$ in time $O(\text{nnz}(X))$ by making a single pass over the data, effectively reducing our dataset to the collection of one-dimensional points $x'_1, \ldots, x'_n \in \mathbb{R}$.

2. Run $k$-means++ seeding on $x'_1, \ldots, x'_n$ to obtain $k$ indices $j_1, \ldots, j_k \in [n]$ indicating the chosen centers $x'_{j_1}, \ldots, x'_{j_k}$ and an assignment $\sigma : [n] \to [k]$ assigning point $x'_i$ to center $x'_{j_{\sigma(i)}}$. Even though $k$-means++ seeding generally takes $O(nk)$ time in one dimension, we give an efficient implementation, leveraging the fact that points are one-dimensional, which runs in $O(n \log n)$ expected time, independent of $k$. A detailed algorithm description is in the appendix.

3. The one-dimensional $k$-means++ algorithm produces a collection of $k$ centers $x_{j_1}, \ldots, x_{j_k}$, as well as the assignment $\sigma$ mapping each point $x_i$ to the center $x_{j_{\sigma(i)}}$. For each $\ell \in [k]$, we update the cluster center for cluster $\ell$ to be the center of mass of all points assigned to $x_{j_\ell}$.

While the algorithm is straightforward, the main technical difficulty lies in the analysis. In particular, our analysis (1) bounds the approximation loss incurred from the one-dimensional projection in Step 1 and (2) shows that we can implement Step 2 in $O(n \log n)$ expected time, as opposed to $O(nk)$ time. We summarize the theoretical contributions in the following theorems.

**Theorem 2.1.** *The algorithm* PRONE *has expected running time* $O(\text{nnz}(X) + n \log n)$ *on any dataset* $X = \{x_1, \ldots, x_n\} \subset \mathbb{R}^d$. *Moreover, for any* $\delta \in (0, 1/2)$ *and any dataset* $X$, *with probability at least* $1 - \delta$, *the algorithm runs in time* $O(\text{nnz}(X) + n \log(n/\delta))$.

**Theorem 2.2.** *The algorithm* PRONE *achieves an* $\widetilde{O}(k^4)$ *approximation ratio for the* $k$-means objective *with probability at least* 0.9.

To our knowledge, PRONE is the first algorithm for $k$-means running in time $O(nd + n \log n)$ for arbitrary $k$. As mentioned in the paragraph on improving the competitive ratio, we obtain the following corollary of the previous two theorems using a two-stage approach with a coreset:

**Corollary 2.3.** *By using* PRONE *as the* $\alpha$-approximation algorithm in Theorem A.2 and running $k$-means++ *on the resulting coreset, we obtain an algorithm with an approximation ratio of* $O(\log k)$ *that runs in time* $O(\text{nnz}(X) + n \log n + \text{poly}(kd) \log n)$, *with constant success probability.*

Due to space constraints, all proofs of our theoretical results are deferred to the appendix, where we also generalize them beyond $k$-means to clustering objectives that sum up the $z$-th power of Euclidean distances for general $z \geq 1$. The following subsections give a high-level overview of the main techniques we develop to prove our main theorems above.

### 2.1 Efficient Seeding in One Dimension

The $k$-means++ seeding procedure has $k$ iterations, where a new center is sampled in each iteration. Since a new center may need to update $\Omega(n)$ distances to maintain the $D^2$ distribution, which samples each point with probability proportional to its distance to its closest center, a naive analysis leads to a running time of $O(nk)$. A key ingredient in the proof of Theorem 2.1 is showing that, for one-dimensional datasets, $k$-means++ only needs to make $O(n \log n)$ updates, irrespective of $k$.

**Lemma 2.4.** *The* $k$-means++ *seeding procedure can be implemented in expected time* $O(n \log n)$ *in one dimension. Moreover, for any* $\delta \in (0, 1/2)$, *with probability at least* $1 - \delta$, *the implementation runs in time* $O(n \log(n/\delta))$.

The intuition of the proof is as follows: Since points are one-dimensional, we always maintain them in sorted order. In addition, each data point $x_i$ will maintain its center assignment and distance $p_i$ to the closest center. By building a binary tree over the sorted points (where internal nodes maintain sums of $p_i^2$'s), it is easy to sample a new center from the $D^2$ distribution in $O(\log n)$ time. The difficulty is

that adding a new center may result in changes to $p_i$'s of multiple points $x_i$, so the challenge is to bound the number of times these values are updated (see Figure 1 below).

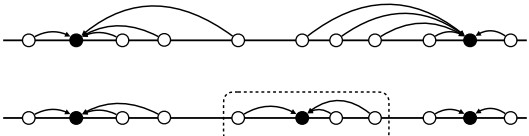

Figure 1: From the top to the bottom, a new center (black circle) is chosen. Every point has an arrow pointing to its closest center. The points in the dashed box are the ones that require updates.

To bound the total running time, we leverage the one-dimensional structure. Observe that, for a new center, the updated points lie in a contiguous interval around the newly chosen center. Once a center is chosen, the algorithm scans the points (to the left and the right) until we reach a point that does not need to be updated. This point identifies that points to the other side of it need not be updated, so we can get away without necessarily checking all $n$ points (see Figure 1). Somewhat surprisingly, when sampling centers from the $D^2$-distribution, the expected number of times that each point will be updated is only $O(\log n)$, which implies a bound of $O(n \log n)$ on the total number of updates in expectation. The analysis of the fact that each point is updated $O(\log n)$ times is non-trivial and uses a carefully designed potential function (Lemma C.5).

## 2.2 Approximation Guarantees from One-Dimensional Projections

Our proof of Theorem 2.2 builds on a line of work studying randomized dimension reduction for clustering problems [14, 21, 11, 53]. Prior work studied randomized dimension reduction for accurate $(1 \pm \epsilon)$-approximations. Our perspective is slightly different; we restrict ourselves to one-dimensional projections and give an upper bound on the distortion.

For any dataset $x_1, \ldots, x_n \in \mathbb{R}^d$, a projection to a random lower-dimensional space affects the pairwise distance between the projected points in a predictable manner — the Johnson-Lindenstrauss lemma which projects to $O(\log n)$ dimensions being a prime example of this fact. When projecting to just one dimension, however, pairwise distances will be significantly affected (by up to $\mathrm{poly}(n)$-factors). Thus, a naive analysis will give a $\mathrm{poly}(n)$-approximation for $k$-means. To improve a $c$-approximation to a $O(\log k)$-approximation, one needs a coreset of size roughly $\mathrm{poly}(c/\log k)$. This bound becomes vacuous when $c$ is polynomial in $n$ since there are at most $n$ dataset points.

However, although many pairwise distances are significantly distorted, we show that the $k$-means cost is only affected by a $\mathrm{poly}(k)$-factor. At a high level, this occurs because the $k$-means cost optimizes a sum of pairwise distances (according to a chosen clustering). The individual summands, given by pairwise distances, will change significantly, but the overall sum does not. Our proof follows the approach of [21], which showed that (roughly speaking) pairwise distortion of the $k$ optimal centers suffices to argue about the $k$-means cost. The $k$ optimal centers will incur maximal pairwise distortion $\mathrm{poly}(k)$ when projected to one dimension (because there are only $O(k^2)$ pairwise distances among the $k$ centers). This allows us to lift an $r$-approximate solution after the projection to an $O(k^4 r)$-approximate solution for the original points.

**Lemma 2.5** (Informal). *For any set $X$ of points in $\mathbb{R}^d$, the following occurs with probability at least 0.9 over the choice of a standard Gaussian vector $v \in \mathbb{R}^d$. Letting $X' \subset \mathbb{R}$ be the one-dimensional projection of $X$ onto $v$, any $r$-approximate $k$-means clustering of $X'$ gives an $O(k^4 r)$-approximate clustering of $X$ with the same clustering partition.*

## 3 Experimental Results

In this section, we outline the experimental evaluation of our algorithm. The experiments evaluate the algorithms in two different ways. For each, we measure the running time and the $k$-means cost of the resulting solution (the sum of squares of point-to-center-assigned distances). (1) First, we evaluate our algorithm as part of a pipeline incorporating a coreset construction – the expected use case for our algorithm. (2) Second, we evaluate our algorithm by itself for approximate k-means clustering and compare it to k-means++ [6]. As per Theorems 2.1 and 2.2, we expect our algorithm to be much faster but output an assignment of higher cost. Our goal is to quantify these differences empirically.

All experiments were run on Linux using a notebook with a 3.9 GHz 12th generation Intel Core i7 six-core processor and 32 GiB of RAM. All algorithms were implemented in C++, using the `blaze` library for matrix and vector operations performed on the dataset unless specified differently below. We provide our code in the supplementary material for this submission.

**Datasets.** For our experiments, we use the following four datasets:

**KDD** [47]: Training data for the 2004 KDD challenge on protein homology. The dataset consists of $145751$ observations with $77$ real-valued features.

**Song** [12]: Timbre information for $515345$ songs with $90$ features each, used for year prediction.

**Census** [32]: 1990 US census data with $2458285$ observations, each with $68$ categorical features.

**Gaussian**: A synthetic dataset consisting of $240005$ points of dimension $4$. The points are generated by placing a standard normal distribution at a large positive distance from the origin on each axis and sampling $30000$ points. The points are then mirrored so the center of mass remains at the origin. Finally, 5 points are placed on the origin. This is an adversarial example for lightweight coresets [10], which are unlikely to sample points close to the mean of the dataset.

### 3.1  Coreset Construction Comparison

**Experimental Setup.**    Coreset constructions (with multiplicative approximation guarantees) always proceed by first finding an approximate clustering, which constitutes the bulk of the work. The approximate clustering defines a "sensitivity sampling distribution" (we expand on this in the appendix, see also [9]), and a coreset is constructed by repeatedly sampling from the sensitivity sampling distribution. In our first experiment, we evaluate the choice of initial approximation algorithm used to define the sensitivity sampling distribution. We compare the use of $k$-means++ and PRONE. In addition, we also compare the lightweight coresets of [10], which uses the distance to the center of mass as an approximation of the sensitivity sampling distribution. For the remainder of this section, we refer to sensitivity sampling using k-means++ as *Sensitivity* and lightweight coresets as *Lightweight*. All three algorithms produce a coreset, and the experiment will measure the running time of the three algorithms (Table 1) and the quality of the resulting coresets (Figure 2). We implemented all algorithms in C++.

Once a coreset is constructed for each of the algorithms, we evaluate the quality of the coreset by computing the cost of the centers found when clustering the coreset (see Definition A.1). We run a state-of-the-art implementation of Lloyd's $k$-means algorithm from the `scikit-learn` library [57] with the default configuration (repeating 15 times and reporting the mean cost to reduce the variance). The resulting quality of the coresets is compared to a (computationally expensive) *baseline*, which runs k-means++ from the `scikit-learn` library, followed by Lloyd's algorithm with the default configuration on the entire dataset (repeated 5 times to reduce variance).

We evaluate various choices of $k$ ($\{10, 100, 1000\}$) as well as coresets at various relative sizes, $\{0.001, 0.0025, 0.005, 0.01, 0.025, 0.05, 0.1\}$ times the size of the dataset. We use as performance metrics (1) a relative cost, which measures the average cost of the k-means solutions returned by Lloyd's algorithm on each coreset divided by the baseline, and (2) the running time of the coreset construction algorithm.

**Results on Coreset Constructions.**    *Relative cost.*  Figure 2 shows the coreset size ($x$-axis) versus the relative cost ($y$-axis). Each "row" of Figure 2 corresponds to a different value for $k \in \{10, 100, 1000\}$, and each "column" corresponds to a different dataset. Recall that the first three datasets (i.e., the first three columns) are real-world datasets, and the fourth column is the synthetic Gaussian dataset. We note our observations below:

- As expected, on all real-world data sets and all settings of $k$, the relative cost decreases as the coreset size increases.
- In real-world datasets, the specific relative cost of each coreset construction (*Senstivity*, *Lightweight*, and ours) depends on the dataset[4], but roughly speaking, all three share a similar trend. Ours and *Sensitivity* are very close and never more than twice the baseline (usually much better).

---

[4]The spike in relative cost for algorithm *Sensitivity* on the KDD data set for relative size $5 \cdot 10^{-3}$ is due to outliers.

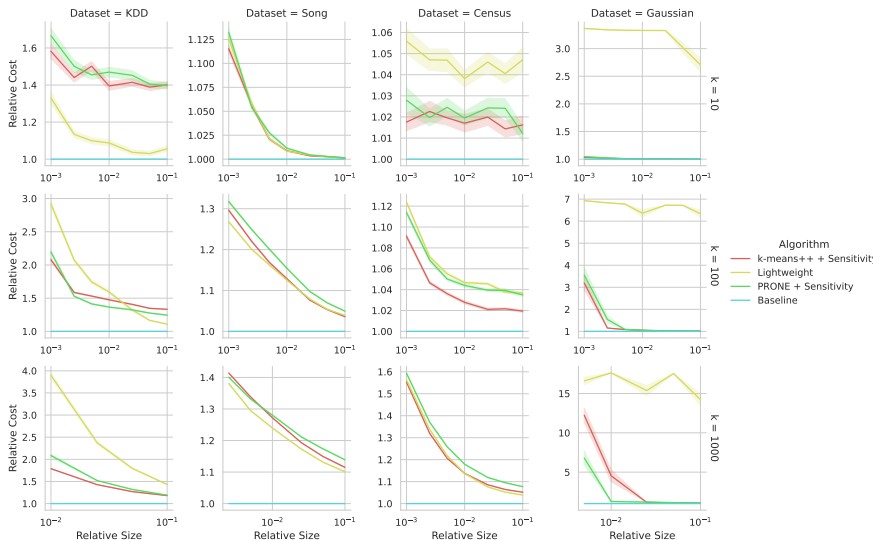

Figure 2: Plot of relative cost versus coreset size on our four datasets. The shaded region indicates standard error. There is no data for relative size values where the coreset size is less than $k$.

- The big difference, distinguishing ours and *Sensitivity* from *Lightweight*, is the fourth column, the synthetic Gaussian dataset. For all settings of $k$, as the coreset size increases, *Lightweight* exhibits a minimal cost decrease and is a factor of 2.7-17x times worse than ours and *Sensitivity* (as well as the baseline). This is expected, as we constructed the synthetic Gaussian dataset to have arbitrarily high cost with *Lightweight*. Due to its multiplicative approximation guarantee, our algorithm does not suffer this degradation. In that sense, our algorithm is more "robust," and achieves worst-case multiplicative approximation guarantees for all datasets.

*Running time.* In (the first table in) Table 1, we show the running time of the coreset construction algorithms as $k$ increases. Notice that as $k$ increases, the relative speedup of our algorithm and *Lightweight* increases in comparison to *Sensitivity*. This is because our algorithm and *Lightweight* have running time which *does not grow with* $k$. In contrast, the running time of *Sensitivity* grows linearly in $k$. In summary, our coreset construction is between **33-192x** faster than *Sensitivity* for large $k$. In addition, our algorithm runs about 3-5x slower than *Lightweight*, depending on the dataset. Our analysis also shows this; both algorithms make an initial pass over the dataset, using $O(nd)$ time, but ours uses an additional $O(n \log n)$ time to process.

### 3.2 Direct k-Means++ Comparison

**Experimental Setup.** This experiment compares our algorithm and $k$-means++ as a stand-alone clustering algorithm, as opposed to as part of a coreset pipeline. We implemented three variants of our algorithm. Each differs in how we sample the random one-dimensional projection. The first is a one-dimensional projection onto a standard Gaussian vector (zero mean and identity covariance). This approach risks collapsing an "important" feature, i.e. a feature with high variance. To mitigate this, we implemented two *data-dependent* variants that use the variance, resp. covariance of the data. Specifically, in the "variance" variant, we use a diagonal covariance matrix, where each entry in the diagonal is set to the empirical variance of the dataset along the corresponding feature. In the "covariance" variant, we use the empirical covariance matrix of the dataset. These variants aim to project along vectors that capture more of the variance of the data than when sampling a vector uniformly at random. Intuitively, the vectors sampled by the biased variants are more correlated with the first principal component of the dataset. For each of our algorithms, we evaluate the $k$-means cost of the output set $C$ of centers when assigning points to the closest center ($\text{cost}_2(X, C)$ in Definition A.1) and when using our algorithm's assignment ($\text{cost}_2(X, C, \sigma)$ defined in Equation (1)).

We evaluated the algorithms for every $k$ in $\{10, 25, 50, 100, 250, 500, 1000, 2500, 5000\}$ and $z = 2$, for solving $k$-means with the $\ell_2$-metric. When evaluating the assignment cost, we ran each of our

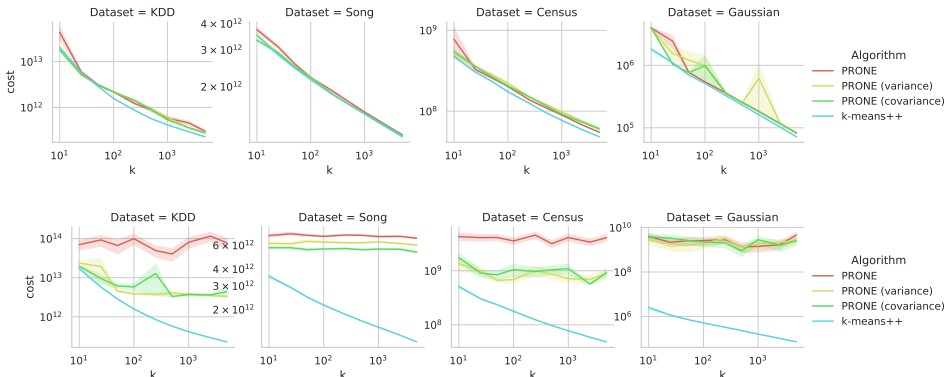

Figure 3: Clustering cost of all our variants compared to $k$-means++. The top row shows the $k$-means cost, and the bottom row shows the cost of the assignment produced by our algorithm.

| Dataset | Algorithm | k 10 | 100 | 1000 |
|---------|-----------|------|-----|------|
| Census | Lightweight | 7.3 | 69.4 | 670.2 |
| | PRONE coreset | 1.5 | 14.1 | 136.3 |
| | Sensitivity | 1.0 | 1.0 | 1.0 |
| Song | Lightweight | 8.9 | 87.2 | 875.3 |
| | PRONE coreset | 2.1 | 19.9 | 187.9 |
| | Sensitivity | 1.0 | 1.0 | 1.0 |
| KDD | Lightweight | 6.4 | 63.0 | 642.8 |
| | PRONE coreset | 2.1 | 19.6 | 192.6 |
| | Sensitivity | 1.0 | 1.0 | 1.0 |
| Gaussian | Lightweight | 2.4 | 17.6 | 174.8 |
| | PRONE coreset | 0.5 | 3.8 | 33.7 |
| | Sensitivity | 1.0 | 1.0 | 1.0 |

| Dataset | Algorithm | k 50 | 500 | 5000 |
|---------|-----------|------|-----|------|
| Census | PRONE | 7.5 | 73.2 | 662.5 |
| | PRONE (variance) | 2.2 | 22.2 | 214.7 |
| | PRONE (covariance) | 1.1 | 10.7 | 117.4 |
| | k-means++ | 1.0 | 1.0 | 1.0 |
| Song | PRONE | 9.7 | 95.5 | 837.5 |
| | PRONE (variance) | 2.3 | 23.1 | 217.2 |
| | PRONE (covariance) | 0.8 | 8.2 | 82.4 |
| | k-means++ | 1.0 | 1.0 | 1.0 |
| KDD | PRONE | 6.9 | 68.3 | 727.5 |
| | PRONE (variance) | 3.1 | 32.0 | 312.4 |
| | PRONE (covariance) | 1.3 | 12.9 | 128.4 |
| | k-means++ | 1.0 | 1.0 | 1.0 |
| Gaussian | PRONE | 1.9 | 18.3 | 165.9 |
| | PRONE (variance) | 2.0 | 17.7 | 162.9 |
| | PRONE (covariance) | 1.7 | 16.1 | 152.6 |
| | k-means++ | 1.0 | 1.0 | 1.0 |

Table 1: Average speedup over sensitivity sampling across all relative sizes for constructing coresets (in the first table) and average speedup over $k$-means++ as a stand-alone clustering algorithm (in the second table). The tables with the full range of parameters can be found in the appendix.

algorithms 100 times for each $k$ and five times when computing the nearest neighbor assignment, and we report the average cost of the solutions and the average running time. Due to lower variance and much higher runtime, k-means++ was run five times.

**Results on Direct $k$-Means++ Comparison.** *Cost.* Figure 3 (on top) shows the cost of the centers found by our algorithm compared to those found by the k-means++ algorithm after computing *the optimal assignment of points to the centers computed by the algorithm* (computing this takes time $O(ndk)$). That is, we compare the values of $\mathsf{cost}_2(X, C)$ in Definition A.1. In summary, the k-means cost of all three variants of our algorithm are roughly the same and closely match that of k-means++. On the Gaussian data set, one run of the biased algorithm failed to pick a center from the cluster at the origin, leading to a high "outlier" cost and a corresponding spike in the plot.

We also compared the k-means cost for the assignment *computed by our algorithm* (so that our algorithm only takes time $O(nd + n \log n)$ and *not* $O(ndk)$) with the cost of k-means++ (bottom row of Figure 3). That is, we compare the values of $\mathsf{cost}_2(X, C, \sigma)$ defined in Equation (1). The clustering cost of our algorithms is higher than that of k-means++. This is the predicted outcome from our theoretical results; recall Theorem 2.2 gives a $\mathrm{poly}(k)$-approximation, as opposed to $O(\log k)$ from $k$-means++. On the real-world data sets, it is between one order of magnitude (for $k = 10$) and two orders of magnitude (for $k = 5000$) worse than k-means++ for our unbiased variant and between a factor 2 (for $k = 10$) and one order of magnitude (for $k = 5000$) worse than k-means++ for our biased and covariance variants.

*Running time.* Table 1 shows the relative running time of our algorithm compared to k-means++, assuming that no nearest-center assignment is computed. Our algorithms are designed to have a running time independent of $k$, so we can see, from the second table in Figure 1, all of our variants offer significant speedups.

- The running time of our algorithm stays almost constant as $k$ increases while the running time of k-means++ scales linearly with $k$. Specifically for $k = 25$, even our slowest variants have about the same running time as k-means++, while for $k = 5000$, it is at least **82x** faster, and our fastest version is up to **837x** faster over k-means++.

- The two variants can affect the quality of the chosen centers by up to an order of magnitude, but they are also significantly slower. The "variance" and "covariance" variants are slower (between 2-4x slower and up to 10x slower, respectively) than the standard variant, and they also become slower as the dimensionality $d$ increases. We believe these methods could be further sped up, as the `blaze` library's variance computation routine appears inefficient for our use case.

## 4  Conclusion and Limitations

To summarize, we present a simple algorithm that provides a new tradeoff between running time and approximation ratio. Our algorithm runs in expected time $O(\mathsf{nnz}(X) + n \log n)$ to produce a $\mathrm{poly}(k)$-approximation; with additional $\mathrm{poly}(kd) \cdot \log n$ time, we improve the approximation to $O(\log k)$. This latter bound matches that of $k$-means++ but offers a significant speedup.

Within a pipeline for constructing coresets, our experiments show that the quality of the coreset produced (when using our algorithm as the initial approximation) outperforms the sensitivity sampling algorithm. It is slower than the lightweight coreset algorithm, but it is more "robust" as it is independent of the diameter of the data set. It does not suffer from the drawback of having an additive error linear in the diameter of the dataset, which can arbitrarily increase the cost of the lightweight coreset algorithm. When computing an optimal assignment for the centers returned by our algorithm, its cost roughly matches the cost for k-means++. When directly using the assignment produced by one variant of our algorithm, its cost is between a factor 2 and 10 worse while being up to 300 times faster.

Our experiments and running time analysis show that our algorithm is very efficient. However, the clustering quality achieved by our algorithm is sometimes not as good as other, slower algorithms. We show that this limitation is insignificant when we use our algorithm to construct coresets. It remains an interesting open problem to understand the best clustering quality (e.g., in terms of approximation ratio) an algorithm can achieve while being as efficient as ours, i.e., running in time $O(nd + n \log n)$. Another interesting problem is whether other means of projecting the dataset into a $O(1)$ dimensional space exist, which lead to algorithms with improved approximation guarantees and running time faster than $O(ndk)$.

## Acknowledgments and Disclosure of Funding

Moses Charikar was supported by a Simons Investigator award.

Lunjia Hu was supported by Moses Charikar's and Omer Reingold's Simons Investigators awards, Omer Reingold's NSF Award IIS-1908774, and the Simons Foundation Collaboration on the Theory of Algorithmic Fairness. Part of this work was done while Erik Waingarten was a postdoc at Stanford University, supported by an NSF postdoctoral fellowship and by Moses Charikar's Simons Investigator Award.

This project has received funding from the European Research Council (ERC) under the European Union's Horizon 2020 research and innovation programme (Grant agreement No. 101019564 "The Design of Modern Fully Dynamic Data Structures (MoDynStruct)" and the Austrian Science Fund (FWF) project Z 422-N, project "Static and Dynamic Hierarchical Graph Decompositions", I 5982-N, and project "Fast Algorithms for a Reactive Network Layer (ReactNet)", P 33775-N, with additional funding from the netidee SCIENCE Stiftung, 2020–2024.

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

# A Problem Setup and Background

In this work, we always consider datasets $X = \{x_1, \ldots, x_n\} \subset \mathbb{R}^d$ of high-dimensional vectors, and we will measure their distance using the Euclidean ($\ell_2$) distance. Below, we define $(k, z)$-clustering. This problem introduces a parameter $z \geq 1$, which measures the *sensitivity to outliers* (as $z$ grows, the clusterings become more sensitive to points furthest from the cluster center). The case of $k$-means corresponds to $z = 2$, but other values of $z$ capture other well-known clustering objectives, like $k$-median (the case of $z = 1$).

**Definition A.1** ($(k, z)$-Clustering). *Consider a dataset $X = \{x_1, \ldots, x_n\} \subset \mathbb{R}^d$, a desired number of clusters $k \in \mathbb{N}$, and a parameter $z \geq 1$. For a set of $k$ centers $C = \{c_1, \ldots, c_k\} \subset \mathbb{R}^d$, let $\mathsf{cost}_z(X, C)$ denote the cost of using the center set $C$ to cluster $X$, i.e.,*

$$\mathsf{cost}_z(X, C) = \sum_{i=1}^{n} \min_{j \in [k]} \|x_i - c_j\|_2^z.$$

*We let $\mathsf{opt}_{k,z}(X)$ denote the optimal cost over all choices of $C = \{c_1, \ldots, c_k\} \subset \mathbb{R}^d$:*

$$\mathsf{opt}_{k,z}(X) = \inf_{\substack{C \subset \mathbb{R}^d \\ |C| \leq k}} \mathsf{cost}_z(X, C).$$

A $(k, z)$-clustering algorithm has the following specifications. The algorithm receives as input a dataset $X = \{x_1, \ldots, x_n\} \subset \mathbb{R}^d$, as well as two parameters $k \in \mathbb{N}$ and $z \geq 1$. After it executes, the algorithm should output a set of $k$ centers $C = \{c_1, \ldots, c_k\} \subset \mathbb{R}^d$ as well as an assignment $\sigma : [n] \to [k]$ mapping each point $x_i$ to a center $c_{\sigma(i)}$.

We measure the quality of the solution $(C, \sigma)$ using the ratio between its $(k, z)$-clustering cost $\mathsf{cost}_z(X, C, \sigma)$ and the optimal cost $\mathsf{opt}_{k,z}(X)$, where

$$\mathsf{cost}_z(X, C, \sigma) := \sum_{i=1}^{n} \|x_i - c_{\sigma(x_i)}\|_2^z. \tag{1}$$

For any $\mathsf{D} > 1$, an algorithm that produces a $\mathsf{D}$-approximation to $(k, z)$-clustering should guarantee that $\mathsf{cost}_z(X, C, \sigma)$ is at most $\mathsf{D} \cdot \mathsf{opt}_{k,z}(X)$. For a randomized algorithm, the guarantee should hold with large probability (referred to as the success probability) for any input dataset.

## A.1 $k$-Means++ Seeding

The $k$-means++ seeding algorithm is a well-studied algorithm introduced in [6], and it is an important component of our algorithm. Below, we describe it for general $z \geq 1$, not necessarily $z = 2$.

**Definition A.2** ($k$-means++ seeding, for arbitrary $z \geq 1$). *Given $n$ data points $x_1, \ldots, x_n \in \mathbb{R}^d$, the $k$-means++ seeding algorithm produces a set of $k$ centers, $x_{\ell_1}, \ldots, x_{\ell_k}$ with the following procedure:*

1. *Choose $\ell_1$ uniformly at random from $[n]$.*

2. *For $t = 2, \ldots, k$, sample $\ell_t$ as follows. For every $i \in [n]$, let $p_i$ denote the Euclidean distance from $x_i$ to its closest point among $x_{\ell_1}, \ldots, x_{\ell_{t-1}}$. Sample $\ell_t$ from $[n]$ so that the probability $\Pr[\ell_t = i]$ is proportional to $p_i^z$ for every $i \in [n]$. That is,*

$$\Pr[\ell_t = i] = \frac{p_i^z}{\sum_{i' \in [n]} p_i^z}.$$

   *In the context of $k$-means (i.e., when $z = 2$), the distribution is known as the $D^2$ distribution.*

3. *Output $x_{\ell_1}, \ldots, x_{\ell_k}$.*

In the above description, Step 2 of $k$-means++ needs to maintain, for each dataset point $x_i$, the Euclidean distance $p_i$ to its closest center among the centers selected before the current iteration. This step is implemented by making an entire pass over the dataset for each of the $k - 1$ iterations of Step 2, leading to an $O(ndk)$ running time.

**Theorem A.1** ([6], Theorem 3 in [70]). *For $x_1, \ldots, x_n \in \mathbb{R}^d$, let $X = \{x_1, \ldots, x_n\}$ be the input to the $k$-means++ seeding algorithm. For the output $x_{\ell_1}, \ldots, x_{\ell_k}$ of the $k$-means++ seeding algorithm, define $C = \{x_{\ell_1}, \ldots, x_{\ell_k}\}$. Then*

$$\mathbb{E}[\mathsf{cost}_z(X, C)] = O(2^{2z} \log k) \cdot \mathsf{opt}_{k,z}(X).$$

## A.2 Coresets via Sensitivity Sampling

One of our algorithm's applications is constructing coresets for $(k, z)$-clustering. We give a formal definition and describe the primary technique for building coresets.

**Definition A.3.** *Given a dataset $X = \{x_1, \ldots, x_n\} \subset \mathbb{R}^d$, as well as parameters $k \in \mathbb{N}$, $z \geq 1$ and $\varepsilon > 0$, a (strong) $\varepsilon$-coreset for $(k, z)$-clustering is specified by a set of points $Y \subset \mathbb{R}^d$ and a weight function $w \colon Y \to \mathbb{R}_{\geq 0}$, such that, for every set $C = \{c_1, \ldots, c_k\} \subset \mathbb{R}^d$,*

$$(1 - \varepsilon) \cdot \mathsf{cost}_z(X, C) \leq \sum_{y \in Y} w(y) \cdot \min_{j \in [k]} \|y - c_j\|_2^z \leq (1 + \varepsilon) \cdot \mathsf{cost}_z(X, C).$$

Coresets are constructed via "sensitivity sampling," a technique that, given an approximate clustering of a dataset $X$, produces a probability distribution such that sampling enough points from this distribution results in a coreset.

**Definition A.4** (Sensitivity Sampling). *Consider a dataset $X = \{x_1, \ldots, x_n\} \subset \mathbb{R}^d$, as well as parameters $k \in \mathbb{N}$, $z \geq 1$. For a centet set $C = \{c_1, \ldots, c_k\} \subset \mathbb{R}^d$ and assignment $\sigma \colon [n] \to [k]$, let $X_j = \{x_i : \sigma(i) = j\}$. We let $\mathcal{D}$ be a distribution supported on $X$ where*

$$\Pr_{\boldsymbol{x} \sim \mathcal{D}}[\boldsymbol{x} = x_i] \propto \frac{\|x_i - c_{\sigma(i)}\|_2^z}{\sum_{j=1}^n \|x_j - c_{\sigma(j)}\|_2^z} + \frac{1}{|X_{\sigma(i)}|}.$$

The main theorem that we will use is given below, which shows that given a center set and an assignment that gives an $\alpha$-approximation to $(k, z)$-clustering, one may sample from the distribution $\mathcal{D}$ defined about in order to generate a coreset with high probability.

**Theorem A.2** ([15]). *For any dataset $X = \{x_1, \ldots, x_n\} \subset \mathbb{R}^d$ and any parameters $k \in \mathbb{N}$ and $z \geq 1$, suppose that $C = \{c_1, \ldots, c_k\} \subset \mathbb{R}^d$ and $\sigma \colon [n] \to [k]$ is a $\alpha$-approximation to $(k, z)$-clustering, i.e.,*

$$\sum_{i=1}^n \|x_i - c_{\sigma(i)}\|_2^z \leq \alpha \, \mathsf{opt}_{k,z}(X).$$

*Letting $\mathcal{D}$ denote the distribution specified in Definition A.4, the following occurs with high probability.*

- *We let $\boldsymbol{y}_1, \ldots, \boldsymbol{y}_s$ denote independent samples from $\mathcal{D}$, and $w(\boldsymbol{y}_i)$ be the inverse of the probability that $\boldsymbol{y}_i$ is sampled according to $\mathcal{D}$. We set $s \geq \mathrm{poly}(kd \cdot \alpha \cdot 2^z / \varepsilon)$.*

- *The set $\boldsymbol{Y} = \{y_1, \ldots, y_s\}$ with weights $w$ is an $\varepsilon$-coreset for $(k, z)$-clustering.*

## A.3 A Simple Lemma

We will repeatedly use the following simple lemma.

**Lemma A.3.** *Let $a, b \in \mathbb{R}_{\geq 0}$ be any two numbers and $z \geq 1$. Then, $(a + b)^z \leq 2^{z-1} a^z + 2^{z-1} b^z$.*

*Proof.* The function $\phi(t) = t^z$ is convex for $z \geq 1$, so Jensen's inequality implies $\phi((a + b)/2) \leq (1/2)\phi(a) + (1/2)\phi(b)$. $\qquad\square$

# B  Approximation Guarantees from One-Dimensional Projections

In this section, we prove Theorem 2.2 (rather, the generalization of Theorem 2.2 to any $z \geq 1$) by analyzing the random one-dimensional projection step in our algorithm. In order to introduce some notation, let $X = \{x_1, \ldots, x_n\} \subset \mathbb{R}^d$ be a set of points, and for a partition of $X$ into $k$ sets, $(Y_1, \ldots, Y_k)$, we let the $(k, z)$-*clustering cost of $X$ with the partition* $(Y_1, \ldots, Y_k)$ be

$$\mathsf{cost}_z(Y_1, \ldots, Y_k) = \sum_{i=1}^k \min_{c \in \mathbb{R}^d} \sum_{x \in Y_i} \|x - c\|_2^z \tag{2}$$

and call the $k$ points $c$ selected as minima a set of centers realizing the $(k, z)$-clustering cost of $(Y_1, \ldots, Y_k)$. We note that (2) is a cost function for $(k, z)$-clustering, but it is different from

Definition A.1. In Definition A.1, the emphasis is on the set of $k$ centers $C = \{c_1, \ldots, c_k\}$, and the induced set of clustering of $X$, i.e., the partition $(Y_1, \ldots, Y_k)$ given by assigning points to the closest center, is only implicitly specified by the set of centers. On the other hand, (2) emphasizes the clustering $(Y_1, \ldots, Y_k)$, and the set of $k$ centers implicitly specified by $(Y_1, \ldots, Y_k)$. The optimal set of centers and the optimal clustering will achieve the same cost; however, our proof will mostly consider the clustering $(Y_1, \ldots, Y_k)$ as the object to optimize. Shortly, we will sample a (random) dimensionality reduction map $\mathbf{\Pi} \colon \mathbb{R}^d \to \mathbb{R}^t$ and seek bounds for $t = 1$. We will write $\mathrm{cost}_z(\mathbf{\Pi}(Y_1), \ldots, \mathbf{\Pi}(Y_k))$ for the cost of clustering the points after applying the dimensionality reduction map $\mathbf{\Pi}$ to the partition $Y_1, \ldots, Y_k$. Namely, we write

$$\mathrm{cost}_z(\mathbf{\Pi}(Y_1), \ldots, \mathbf{\Pi}(Y_k)) = \sum_{i=1}^{k} \min_{c \in \mathbb{R}^t} \sum_{x \in Y_i} \|\mathbf{\Pi}(x) - c\|_2^z.$$

**Definition B.1.** *For a set of points $X = \{x_1, \ldots, x_n\} \subset \mathbb{R}^d$, we use $X_1^*, \ldots, X_k^*$ of $X$ to denote the partition of $X$ with minimum $(k, z)$-clustering cost and we use $C^* = \{c_1^*, \ldots, c_k^*\} \subset \mathbb{R}^d$ to denote a set of $k$ centers which realizes the $(k, z)$-clustering cost of $X_1^*, \ldots, X_k^*$, i.e., the set of centers which satisfies*

$$\mathrm{cost}_z(X_1^*, \ldots, X_k^*) = \sum_{i=1}^{k} \sum_{x \in X_i^*} \|x - c_i^*\|_2^z.$$

*By slight abuse of notation, we also let $c^* \colon X \to C^*$ be the map which sends every point of $X$ to its corresponding center (i.e., if $x \in X_i^*$, then $c^*(x)$ is the point $c_i^*$).*

We prove the following lemma, which generalizes Lemma 2.5 from $k$-means to $(k, z)$-clustering (recall that $k$-means corresponds to the case of $z = 2$).

**Lemma B.1** (Effect of One-Dimensional Projection on $(k, z)$-Clustering). *For $n, d, k \in \mathbb{N}$ and $z \geq 1$, let $X = \{x_1, \ldots, x_n\} \subset \mathbb{R}^d$ be an arbitrary dataset. We consider the (random) linear map $\mathbf{\Pi} \colon \mathbb{R}^d \to \mathbb{R}$ given by sampling $\mathbf{g} \sim \mathcal{N}(0, I_d)$ and setting*

$$\mathbf{\Pi}(x) = \langle x, \mathbf{g} \rangle.$$

*With probability at least $0.9$ over $\mathbf{g}$, the following occurs:*

- *We consider the projected dataset $\boldsymbol{X}' = \{\boldsymbol{x}'_1, \ldots, \boldsymbol{x}'_n\} \subset \mathbb{R}$ be given by $\boldsymbol{x}'_i = \mathbf{\Pi}(x_i)$, and*

- *For any $r \geq 1$, we let $(Y_1, \ldots Y_k)$ denote any partition of $X$ satisfying*

$$\mathrm{cost}_z(\mathbf{\Pi}(Y_1), \ldots, \mathbf{\Pi}(Y_k)) \leq r \cdot \min_{c_1, \ldots, c_k \in \mathbb{R}} \sum_{i=1}^{n} \min_{j \in [k]} |\boldsymbol{x}'_i - c_j|^z.$$

*Then,*

$$\mathrm{cost}_z(Y_1, \ldots, Y_k) \leq 2^{O(z)} \cdot k^{2z} \cdot r \cdot \mathrm{opt}_{k,z}(X).$$

By setting $z = 2$, we obtain the desired bound from Lemma 2.5. We can immediately see that, from Lemma B.1, and the approximation guarantees of $k$-means++ (or rather, its generalization to $z \geq 1$) in Theorem A.1, we obtain our desired approximation guarantees. Below, we state the generalization of Theorem 2.2 to all $z \geq 1$ and, assuming Lemma B.1, its proof.

**Theorem B.2** (Generalization of Theorem 2.2 to $z \geq 1$). *For $n, d, k \in \mathbb{N}$ and $z \geq 1$, let $X = \{x_1, \ldots, x_n\} \subset \mathbb{R}^d$ be an arbitrary dataset. We consider the following generalization of our algorithm* PRONE:

1. *Sample a random Gaussian vector $\mathbf{g} \sim \mathcal{N}(0, I_d)$ and consider the projection $\boldsymbol{X}' = \{\boldsymbol{x}'_1, \ldots, \boldsymbol{x}'_n\}$ given by $\boldsymbol{x}'_i = \mathbf{\Pi}(x_i)$, for $\mathbf{\Pi}(x) = \langle x, \mathbf{g} \rangle \in \mathbb{R}$.*

2. *Execute the (generalization of the) $k$-means++ seeding strategy for $z \geq 1$ of Definition A.2 with the dataset $\boldsymbol{X}' \subset \mathbb{R}$, and let $\boldsymbol{x}'_{j_1}, \ldots, \boldsymbol{x}'_{j_k} \in \mathbb{R}$ denote the centers and $(\boldsymbol{Y}_1, \ldots, \boldsymbol{Y}_k)$ denote the partition of $X$ specifying the $k$ clusters found.*

*3. Output the clustering $(\boldsymbol{Y}_1, \ldots, \boldsymbol{Y}_k)$, and the set of centers $\boldsymbol{c}_1, \ldots, \boldsymbol{c}_k \in \mathbb{R}^d$ where*

$$\boldsymbol{c}_\ell = \mathop{\mathbb{E}}_{\boldsymbol{x} \sim \boldsymbol{Y}_\ell} [\boldsymbol{x}] \in \mathbb{R}^d.$$

*Then, with probability at least $0.8$ over the execution of the algorithm,*

$$\mathsf{cost}_z(\boldsymbol{Y}_1, \ldots, \boldsymbol{Y}_k) \leq \sum_{\ell=1}^k \sum_{x \in Y_\ell} \|x - \boldsymbol{c}_\ell\|_2^z \leq 2^{O(z)} \cdot k^{2z} \cdot \log k \cdot \mathsf{opt}_{k,z}(X).$$

*Proof of Theorem B.2 assuming Lemma B.1.* We consider the case (over the randomness in the execution of the algorithm) that:

1. The conclusions of Lemma B.1 hold for the projected dataset $\boldsymbol{X}'$ (which happens with probability at least $0.9$) by Lemma B.1.

2. The execution of the generalization $k$-means++ seeding strategy on $\boldsymbol{X}'$ (from Definition A.2) produces a set of centers $\{\boldsymbol{x}'_{j_1}, \ldots, \boldsymbol{x}'_{j_k}\} \subset \mathbb{R}$ which cluster $\boldsymbol{X}'$ with cost at most $O(2^{2z} \log k) \cdot \mathsf{opt}_{k,z}(\boldsymbol{X}')$ (which also happens with probability $0.9$ by Markov's inequality).

By a union bound, both hold with probability at least $0.8$. We now use Lemma B.1 to upper bound the cost of the clustering $(\boldsymbol{Y}_1, \ldots, \boldsymbol{Y}_k)$. The first inequality is trivial; suppose we let $\hat{c}_1^*, \ldots, \hat{c}_\ell^* \in \mathbb{R}^d$ be the centers which minimize for each $\ell \in [k]$

$$\min_{\hat{c}_\ell \in \mathbb{R}^d} \sum_{x \in \boldsymbol{Y}_\ell} \|x - \hat{c}_\ell\|_2^z = \sum_{x \in \boldsymbol{Y}_\ell} \|x - \hat{c}_\ell^*\|_2^z.$$

Then, we trivially have

$$\mathsf{cost}_z(\boldsymbol{Y}_1, \ldots, \boldsymbol{Y}_k) = \sum_{\ell=1}^k \sum_{x \in \boldsymbol{Y}_\ell} \|x - \hat{c}_\ell^*\|_2^z \leq \sum_{\ell=1}^k \sum_{x \in \boldsymbol{Y}_\ell} \|x - \boldsymbol{c}_\ell\|_2^z.$$

Furthermore, we can also show a corresponding upper bound. For each $\ell \in [k]$, recall that $\boldsymbol{c}_\ell \in \mathbb{R}^d$ is the center of mass of $\boldsymbol{Y}_k$, so we can apply the triangle inequality and Lemma A.3

$$\sum_{x \in \boldsymbol{Y}_\ell} \|x - \boldsymbol{c}_\ell\|_2^z \leq 2^{z-1} \sum_{x \in \boldsymbol{Y}_\ell} \|x - \hat{c}_\ell^*\|_2^z + 2^{z-1} |\boldsymbol{Y}_\ell| \cdot \|\hat{c}_\ell^* - \mathop{\mathbb{E}}_{\boldsymbol{x} \sim \boldsymbol{Y}_\ell} [\boldsymbol{x}]\|_2^z$$

$$\leq 2^{z-1} \sum_{x \in \boldsymbol{Y}_\ell} \|x - \hat{c}_\ell^*\|_2^z + 2^{z-1} |\boldsymbol{Y}_\ell| \cdot \mathop{\mathbb{E}}_{\boldsymbol{x} \sim \boldsymbol{Y}_\ell} [\|x - \hat{c}_\ell^*\|_2^z],$$

where the second inequality is Jensen's inequality, since $\phi(x) = \|\hat{c}_\ell^* - x\|_2^z$ is convex for $z \geq 1$. Thus, we have upper-bounded

$$\sum_{x \in \boldsymbol{Y}_\ell} \|x - \boldsymbol{c}_\ell\|_2^z \leq 2^z \sum_{x \in \boldsymbol{Y}_\ell} \|x - \hat{c}_\ell^*\|_2^z,$$

and therefore

$$\sum_{\ell=1}^k \sum_{x \in \boldsymbol{Y}_\ell} \|x - \boldsymbol{c}_\ell\|_2^z \leq 2^z \cdot \mathsf{cost}_z(\boldsymbol{Y}_1, \ldots, \boldsymbol{Y}_k). \tag{3}$$

The final step involves relating $\mathsf{cost}_z(\boldsymbol{Y}_1, \ldots, \boldsymbol{Y}_k)$ using the conclusions of Lemma B.1. Notice that our algorithm produces the clustering $(\boldsymbol{Y}_1, \ldots, \boldsymbol{Y}_k)$ of $\boldsymbol{X}'$ which is specified by letting

$$\boldsymbol{Y}_\ell = \left\{ x_i \in X : \forall j' \in [k], |\boldsymbol{x}'_i - \boldsymbol{x}'_{j_\ell}| \leq |\boldsymbol{x}'_i - \boldsymbol{x}_{j'}|^z \right\},$$

and by the event (2), we have $\mathsf{cost}_z(\boldsymbol{\Pi}(\boldsymbol{Y}_1), \ldots, \boldsymbol{\Pi}(\boldsymbol{Y}_k)) \leq O(2^{2z} \log k) \cdot \mathsf{opt}_{k,z}(\boldsymbol{X}')$. By event (1), Lemma B.1 implies that $\mathsf{cost}_z(\boldsymbol{Y}_1, \ldots, \boldsymbol{Y}_k) \leq 2^{O(z)} \cdot k^{2z} \cdot O(2^{2z} \log k) \cdot \mathsf{opt}_{k,z}(X)$. Combined with (3), we obtain our desired bound. $\qquad\square$

## B.1 Proof of Lemma B.1

We now turn to the proof of Lemma B.1, where our analysis will proceed in two steps. First, we assume a fixed dimensionality reduction map $\Pi \colon \mathbb{R}^d \to \mathbb{R}^t$, which satisfies two geometrical conditions on $\Pi$. Under these conditions, we show how to "lift" an approximate clustering of the mapped points in $\mathbb{R}^t$ to an approximate clustering of the original dataset in $\mathbb{R}^d$ at the cost of weakening the approximation ratio. Then, we show that a simple one-dimensional projection $\mathbf{\Pi} \colon \mathbb{R}^d \to \mathbb{R}$ given by $\mathbf{\Pi}(x) = \langle x, \mathbf{g} \rangle$, for $\mathbf{g}$ being sampled from a $d$-dimensional standard Gaussian, satisfies the geometrical conditions of our lemma.

**Lemma B.3.** *Let $X = \{x_1, \ldots, x_n\} \subset \mathbb{R}^d$ and $\Pi \colon \mathbb{R}^d \to \mathbb{R}^t$ be a linear map. Let $C = \{c_1^*, \ldots, c_k^*\} \subset \mathbb{R}^d$ denote the set of centers minimizing $\mathsf{cost}_z(X, C)$, and $(X_1^*, \ldots, X_k^*)$ denote the optimal $(k, z)$-clustering, and suppose that for the parameters $\mathsf{D}_1, \mathsf{D}_2, \mathsf{D}_3 \geq 1$, the following conditions hold:*

- **Centers Don't Contract**: *Every $i, j \in [k]$ satisfies*

$$\|c_i^* - c_j^*\|_2 \leq \mathsf{D}_1 \cdot \|\Pi(c_i^*) - \Pi(c_j^*)\|_2.$$

- **Cost of** $(X_1^*, \ldots, X_k^*)$ **does not Increase**: *We have that*

$$\sum_{i=1}^k \sum_{x \in X_i^*} \|\Pi(x) - \Pi(c_i^*)\|_2^z \leq \mathsf{D}_2 \cdot \mathsf{cost}_z(X_1^*, \ldots, X_k^*).$$

- **Approximately Optimal** $(\Pi(Y_1), \ldots, \Pi(Y_k))$: *The partition $(Y_1, \ldots, Y_k)$ of $X$ is $\mathsf{D}_3$- approximately optimal for $\Pi(X)$, i.e.,*

$$\mathsf{cost}_z(\Pi(Y_1), \ldots, \Pi(Y_k)) \leq \mathsf{D}_3 \cdot \min_{c_1, \ldots, c_k \in \mathbb{R}^t} \sum_{x \in X} \min_{j \in [k]} \|\Pi(x) - c_j\|_2^z.$$

*Then,*
$$\mathsf{cost}_z(Y_1, \ldots, Y_k) \leq \left( 2^{z-1} + 2^{3z-2} \mathsf{D}_1^z \mathsf{D}_2 (1 + \mathsf{D}_3) \right) \cdot \mathsf{cost}_z(X_1^*, \ldots, X_k^*).$$

Before starting the proof of Lemma B.3, we show that projecting points onto a random Gaussian vector gives the first two desired guarantees of the above lemma with $\mathsf{D}_1 := (k^2/\delta)$ and $\mathsf{D}_2 := 2^{O(z)}/\delta$ with probability at least $1 - \delta$. The first lemma that we state below shows that the first condition of Lemma B.3 is satisfied with high probability, and the second lemma that the second condition of Lemma B.3 is satisfied with high probability.

**Lemma B.4** (Centers Don't Contract). *Let $C = \{c_1, \ldots, c_k\} \subset \mathbb{R}^d$ denote any collection of $k$ points and let $\mathbf{\Pi} \colon \mathbb{R}^d \to \mathbb{R}$ be a random map given by*

$$\mathbf{\Pi}(x) = \langle x, \mathbf{g} \rangle$$

*for a randomly chosen vector $\mathbf{g} \sim \mathcal{N}(0, I_d)$. Then, with probability at least $1 - \delta$ over $\mathbf{g}$, every $i, j \in [k]$ satisfies*

$$\left( \frac{\delta}{k^2} \right) \cdot \|c_i - c_j\|_2 \leq \|\mathbf{\Pi}(c_i) - \mathbf{\Pi}(c_j)\|_2.$$

**Lemma B.5** (Cost of $(X_1^*, \ldots, X_k^*)$ does not Increase). *Let $X = \{x_1, \ldots, x_n\} \subset \mathbb{R}^d$ and let $X_1^*, \ldots, X_k^*$ be the partition of $X$, and $c_1^*, \ldots, c_k^* \in \mathbb{R}^d$ be the centers which minimize the $(k, z)$- clustering cost of $X$. Then, with probability a least $1 - \delta$,*

$$\sum_{i=1}^k \sum_{x \in X_i^*} \|\mathbf{\Pi}(x) - \mathbf{\Pi}(c_i^*)\|_2^z \leq \left( 2^{O(z)}/\delta \right) \sum_{i=1}^k \sum_{x \in X_i^*} \|x - c_i^*\|_2^z.$$

*Proof of Lemma B.1 assuming Lemma B.3, Lemma B.4 and Lemma B.5.* We will apply Lemma B.3 by letting $\delta$ be a small enough constant (say, $\delta = 0.01$) to take a union bound. Lemma B.4 implies the first condition with $\mathsf{D}_1 = O(k^2)$ and Lemma B.5 implies the second condition with $\mathsf{D}_2 = 2^{O(z)}$. Finally, the second assumption of Lemma B.1 sets $r = \mathsf{D}_3$, from which we derive the conclusion. $\quad\square$

We now prove Lemma B.4, Lemma B.5. Lemma B.3 is proved in Subsection B.2.

*Proof of Lemma B.4.* The proof relies on the 2-stability property of the Gaussian distribution. Namely, if we let $z \in \mathbb{R}^d$ be an arbitrary vector and we sample a standard Gaussian vector $\mathbf{g} \sim \mathcal{N}(0, I_d)$, the (scalar) random variable $\langle z, \mathbf{g} \rangle$ is distributed like $\|z\|_2 \cdot \mathbf{g}'$, where $\mathbf{g} \sim \mathcal{N}(0, 1)$. Using the 2-stability of the Gaussian distribution for every $i, j \in [k]$, we have that $\|\mathbf{\Pi}(c_i) - \mathbf{\Pi}(c_j)\|_2^2$ is distributed as $(\mathbf{g}')^2 \|c_i - c_j\|_2^2$, where $\mathbf{g}'$ is distributed as a (one-dimensional) Gaussian $\mathcal{N}(0, 1)$. Thus, by a union bound, the probability that there exists a pair $i, j \in [k]$, which satisfies $\|\mathbf{\Pi}(c_i) - \mathbf{\Pi}(c_j)\|_2^2 < \alpha^2 \cdot \|c_i - c_j\|_2^2$ is at most $k^2$ times the probability that a Gaussian random variable lies in $[-\alpha, \alpha]$, and this probability is easily seen to be less than $\alpha$. Setting $\alpha = \delta / k^2$ gives the desired lemma. □

*Proof of Lemma B.5.* Similarly to the proof of Lemma B.4, we have that $\|\mathbf{\Pi}(x) - \mathbf{\Pi}(c_i^*)\|_2^z$ is distributed as $|\mathbf{g}'|^z \cdot \|x - c_i^*\|_2^z$, where $\mathbf{g}'$ is distributed as a (one-dimensional) Gaussian $\mathcal{N}(0, 1)$. By linearity of expectation,

$$\mathbb{E}_{\mathbf{\Pi} \sim \mathcal{J}_d} \left[ \sum_{i=1}^{k} \sum_{x \in X_i^*} \|\mathbf{\Pi}(x) - \mathbf{\Pi}(c_i^*)\|_2^z \right] = \sum_{i=1}^{k} \sum_{x \in X_i^*} \mathbb{E}_{\mathbf{g}' \sim \mathcal{N}(0,1)} [|\mathbf{g}_i'|^z] \cdot \|x - c_i^*\|_2^z.$$

To conclude, note that for $z \geq 1$, there is some $\alpha > 1$ such that $\alpha z$ is an even integer and $\alpha \leq 2$. Thus, by Jensen's inequality and the fact that $f(x) = x^{1/\alpha}$ is concave we can write

$$\mathbb{E}_{\mathbf{g}' \sim \mathcal{N}(0,1)} [|\mathbf{g}'|^z] \leq (\mathbb{E}[(\mathbf{g}')^{\alpha z}])^{1/\alpha}$$

Now note that all odd moments of the Gaussian distribution are zero by symmetry. Thus, for the moment generating function $\mathbb{E}[e^{\mathbf{g}'}]$ it holds that

$$\mathbb{E}[e^{\mathbf{g}'}] = \sum_{k=0}^{\infty} \frac{1}{(2k)!} \cdot \mathbb{E}[(\mathbf{g}')^{2k}].$$

As $\mathbb{E}[e^{\mathbf{g}'}] \leq e^{1/2}$ it follows that

$$(\mathbb{E}[(\mathbf{g}')^{\alpha z}])^{1/\alpha} \leq \left( (\alpha z)! \mathbb{E}[e^{\mathbf{g}'}] \right)^{1/\alpha} \leq \left( (\alpha z)! e^{1/2} \right)^{1/\alpha} \leq 2^{O(z)}.$$

Applying Markov's inequality now completes the proof. □

## B.2 Proof of Lemma B.3

Let $\{\hat{c}_i\}_{i \in [k]}$ be an optimal set of centers for the partition $(Y_1, \ldots, Y_k)$ for the $(k, z)$-clustering problem on $\Pi(X)$, where $\hat{c}_i \in Y_i$. Specifically, the points $\hat{c}_1, \ldots, \hat{c}_k \in \mathbb{R}^t$ are those which minimize

$$\mathsf{cost}(\Pi(Y_1), \ldots, \Pi(Y_k)) \stackrel{\text{def}}{=} \sum_{i=1}^{k} \sum_{x \in Y_i} \|\Pi(x) - \hat{c}_i\|_2^z.$$

To quantize the cost difference between the centers $c^*$ and the centers $\hat{c}$ we analyze the following value. We assume we mapped every point of $X$ to $\Pi c^*(X)$, and we then compute the cost of the partition $Y_1, \ldots, Y_k$ on this set. Formally, we let

$$\mathsf{val}(\Pi, c^*, Y_1, \ldots, Y_k) = \sum_{i=1}^{k} \sum_{j=1}^{k} |Y_i \cap X_j^*| \cdot \|\Pi(c_j^*) - \hat{c}_i\|_2^z.$$

First, we prove the following simple claim.

**Claim B.6.** *There exists a set of centers $c_1', \ldots, c_k'$ (with possible repetitions) which are chosen among the points $\{c_1^*, \ldots, c_k^*\}$ such that*

$$\sum_{i=1}^{k} \sum_{j=1}^{k} |Y_i \cap X_j^*| \cdot \|\Pi(c_j^*) - \Pi(c_i')\|_2^z \leq 2^z \cdot \mathsf{val}(\Pi, c^*, Y_1, \ldots, Y_k).$$

*Proof.* We will prove the claim using the probabilistic method. For every $i \in [k]$, consider the distribution over center $\{c_1^*, \ldots, c_k^*\}$ which samples a center $\boldsymbol{c}_i'$ as

$$\Pr_{\boldsymbol{c}_i'} \left[ \boldsymbol{c}_i' = c_j^* \right] = \frac{|Y_i \cap X_j^*|}{\sum_{\ell=1}^{d} |Y_i \cap X_j^*|}.$$

Then, we upper bound the expected cost of using the centers $\boldsymbol{c}_i'$. Using Lemma A.3,

$$\mathbb{E} \left[ \sum_{i=1}^{k} \sum_{j=1}^{k} |Y_i \cap X_j^*| \cdot \|\Pi(c_j^*) - \Pi(\boldsymbol{c}_i')\|_2^z \right]$$

$$\leq \sum_{i=1}^{k} \sum_{j=1}^{k} |Y_i \cap X_j^*| \cdot \mathbb{E} \left[ \left( \|\Pi(c_j^*) - \hat{c}_i\|_2 + \|\Pi(\boldsymbol{c}_i') - \hat{c}_i\|_2 \right)^z \right]$$

$$\leq 2^{z-1} \sum_{i=1}^{k} \sum_{j=1}^{k} |Y_i \cap X_j^*| \cdot \|\Pi(c_j^*) - \hat{c}_i\|_2^z + 2^{z-1} \sum_{i=1}^{k} \left( \sum_{j=1}^{k} |Y_i \cap X_j^*| \right) \mathbb{E} \left[ \|\Pi(\boldsymbol{c}_i') - \hat{c}_i\|_2^z \right]$$

$$= 2^{z-1} \cdot \text{val}(\Pi, c^*, Y_1, \ldots, Y_k) + 2^{z-1} \sum_{i=1}^{k} \left( \sum_{j=1}^{k} |Y_i \cap X_j^*| \right) \sum_{\ell=1}^{k} \frac{|Y_i \cap X_\ell^*|}{\sum_{j=1}^{k} |Y_i \cap X_j^*|} \cdot \|\Pi(c_\ell^*) - \hat{c}_i\|_2^z$$

$$= 2^z \cdot \text{val}(\Pi, c^*, Y_1, \ldots, Y_k). \hspace{4cm} \square$$

We now upper bound $\text{cost}_z(Y_1, \ldots, Y_k)$ in terms of $\text{cost}_z(X_1^*, \ldots, X_k^*)$. We do this by going through the centers chosen according to Claim B.6. This will allow us to upper bound the cost of clustering with $(Y_1, \ldots, Y_k)$ in terms of the $\text{cost}_z(X_1^*, \ldots, X_k^*)$ as well as clustering cost involving only pairwise distances from $\{c_1^*, \ldots, c_k^*\}$. Then, we relate to distances after applying the map $\Pi$. Specifically, first notice that if we consider the set of centers $c_1', \ldots, c_k'$ chosen from Claim B.6

$$\text{cost}_z(Y_1, \ldots, Y_n) \leq \sum_{i=1}^{k} \sum_{x \in Y_i} \|x - c_i'\|_2^z$$

$$\leq \sum_{i=1}^{k} \sum_{j=1}^{k} \sum_{x \in Y_i \cap X_j^*} \left( \|x - c_j^*\|_2 + \|c_j^* - c_i'\|_2 \right)^z$$

$$\leq 2^{z-1} \cdot \text{cost}_z(X_1^*, \ldots, X_k^*) + 2^{z-1} \sum_{i=1}^{k} \sum_{j=1}^{k} |Y_i \cap X_j^*| \cdot \|c_j^* - c_i'\|_2^z, \quad (4)$$

where the third inequality uses Lemma A.3 once more. Note that the right-most summation of (4) involves distances which are only among $c_1^*, \ldots, c_k^*$, so by the first assumption of the map $\Pi$ and Claim B.6, we may upper bound

$$\sum_{i=1}^{k} \sum_{j=1}^{k} |Y_i \cap X_j^*| \cdot \|c_j^* - c_i'\|_2^z \leq \mathsf{D}_1^z \sum_{i=1}^{k} \sum_{j=1}^{k} |Y_i \cap X_j^*| \cdot \|\Pi(c_j^*) - \Pi(c_i')\|_2^z$$

$$\leq \mathsf{D}_1^z \cdot 2^z \cdot \text{val}(\Pi, c^*, Y_1, \ldots, Y_k). \quad (5)$$

Combining (4) and (5), we may upper bound

$$\text{cost}_z(Y_1, \ldots, Y_n) \leq 2^{z-1} \cdot \text{cost}_z(X_1^*, \ldots, X_k^*) + \mathsf{D}_1^z \cdot 2^{2z-1} \cdot \text{val}(\Pi, c^*, Y_1, \ldots, Y_k)$$

$$= 2^{z-1} \cdot \text{cost}_z(X_1^*, \ldots, X_k^*) + \mathsf{D}_1^z \cdot 2^{2z-1} \sum_{i=1}^{k} \sum_{j=1}^{k} |Y_i \cap X_j^*| \cdot \|\Pi(c_j^*) - \hat{c}_i\|_2^z.$$

$$(6)$$

We continue upper bounding the right-most expression in (6) by applying the triangle inequality:

$$\sum_{i=1}^{k} \sum_{j=1}^{k} |Y_i \cap X_j^*| \cdot \|\Pi(c_j^*) - \hat{c}_i\|_2^z$$

$$\leq 2^{z-1} \sum_{i=1}^{k} \sum_{j=1}^{k} \sum_{x \in Y_i \cap X_j^*} \|\Pi(x) - \Pi(c_j^*)\|_2^z + 2^{z-1} \sum_{i=1}^{k} \sum_{j=1}^{k} \sum_{x \in Y_i \cap X_j^*} \|\Pi(x) - \hat{c}_i\|_2^z$$

$$\leq 2^{z-1} \mathsf{D}_2 \cdot \mathsf{cost}_z(X_1^*, \ldots, X_k^*) + 2^{z-1} \cdot \mathsf{cost}_z(\Pi(Y_1), \ldots, \Pi(Y_k)). \tag{7}$$

By the third assumption of the lemma, we note that

$$\mathsf{cost}_z(\Pi(Y_1), \ldots, \Pi(Y_k)) \leq \mathsf{D}_3 \cdot \min_{c_1, \ldots, c_k \in \mathbb{R}^t} \sum_{x \in X} \min_{j \in [k]} \|\Pi(x) - c_j\|_2^z$$

$$\leq \mathsf{D}_3 \cdot \sum_{j=1}^{k} \sum_{x \in X_j} \|\Pi(x) - \Pi(c_j^*)\|_2^z \leq \mathsf{D}_3 \mathsf{D}_2 \cdot \mathsf{cost}_z(X_1^*, \ldots, X_k^*). \tag{8}$$

Summarizing by plugging (7) and (8) into (6), can upper bound

$$\mathsf{cost}_z(Y_1, \ldots, Y_n) \leq \left(2^{z-1} + 2^{3z-2} \cdot \mathsf{D}_1^z \mathsf{D}_2 (1 + \mathsf{D}_3)\right) \cdot \mathsf{cost}_z(X_1^*, \ldots, X_k^*).$$

## C    Efficient Seeding in One Dimension

In this section, we prove Theorem 2.1, which shows an upper bound for the running time of our algorithm PRONE. As in Theorem B.2, we consider a generalized version of PRONE where we run $k$-means++ seeding for general $z \geq 1$ (Definition A.2) in Step 2. We prove the following generalized version of Theorem 2.1:

**Theorem C.1** (Theorem 2.1 for general $z \geq 1$). *Let $X = \{x_1, \ldots, x_n\} \subset \mathbb{R}^d$ be a dataset consisting of $n$ points in $d$ dimensions. Assume that $d \leq \mathsf{nnz}(X)$, which can be ensured after removing redundant dimensions $j \in [d]$ where the $j$-th coordinate of every $x_i$ is zero. For any $z \geq 1$, the algorithm PRONE (for general $z$ as in Theorem B.2) has expected running time $O(\mathsf{nnz}(X) + 2^{z/2} n \log n)$ on $X$. For any $\delta \in (0, 1/2)$, with probability at least $1 - \delta$, the algorithm runs in time $O(\mathsf{nnz}(X) + 2^{z/2} n \log(n/\delta))$. Moreover, the algorithm always runs in time $O(\mathsf{nnz}(X) + n \log n + nk)$.*

To prove Theorem 2.1, we show an efficient implementation (Algorithm 1) of the $k$-means++ seeding procedure that runs in expected time $O(2^{z/2} n \log n)$ for one-dimensional points (Lemma C.3). A naive implementation of the seeding procedure would take $\Theta(nk)$ time in one dimension because we need $\Theta(n)$ time to update $p_i$ and sample from the $D^2$ distribution to add each of the $k$ centers. To obtain an improved and provable running time, we use a basic binary tree data structure to sample from the $D^2$ distribution more efficiently, and we use a potential argument to bound the number of updates to $p_i$.

The data structure $S$ we use in Algorithm 1 can be implemented as a basic binary tree, as described in more detail in Appendix D. The data structure $S$ keeps track of $n$ nonnegative numbers $s_1, \ldots, s_n$ corresponding to $p_1^z, \ldots, p_n^z$ and it supports the following operations:

1.  $\mathtt{Initialize}(a)$. Given an array $a = (a_1, \ldots, a_n) \in \mathbb{R}_{\geq 0}^n$, the operation $\mathtt{Initialize}(a)$ creates a data structure $S$ that keeps track of the numbers $s_1, \ldots, s_n$ initialized so that $(s_1, \ldots, s_n) = (a_1, \ldots, a_n)$. This operation runs in $O(n)$ time.

2.  $\mathtt{Sum}(S)$. The operation $\mathtt{Sum}(S)$ returns the sum $s_1 + \cdots + s_n$. This operation runs in $O(1)$ time, as the value will be maintained as the data structure is updated.

3.  $\mathtt{Find}(S, r)$. Given a number $r \in [0, \sum_{i=1}^{n} s_i)$, the operation $\mathtt{Find}(S, r)$ returns the unique index $\ell \in \{1, \ldots, n\}$ such that

$$\sum_{i=1}^{\ell-1} s_i \leq r < \sum_{i=1}^{\ell} s_i.$$

    This operation runs in $O(\log n)$ time.

4.  $\mathtt{Update}(S, a, i_1, i_2)$. Given an array $a = (a_1, \ldots, a_n) \in \mathbb{R}_{\geq 0}^n$ and indices $i_1, i_2$ satisfying $1 \leq i_1 \leq i_2 \leq n$, the operation $\mathtt{Update}(S, a, i_1, i_2)$ performs the updates $s_i \leftarrow a_i$ for every $i = i_1, i_1 + 1, \ldots, i_2$. This operation runs in $O((i_2 - i_1 + 1) + \log n)$ time.

**Algorithm 1:** Efficient $k$-means++ seeding in one dimension

---

**Input:** Points $x_1, \ldots, x_n \in \mathbb{R}$; $k \in \mathbb{Z}$ satisfying $1 \leq k \leq n$; real number $z \geq 1$.
**Output:** Centers $x_{\ell_1}, \ldots, x_{\ell_k} \in \mathbb{R}$; assignment $\sigma : [n] \to [k]$.

1 Sort and re-order the points so that $x_1 \leq \cdots \leq x_n$;
2 Choose $\ell_1$ uniformly at random from $\{1, \ldots, n\}$;
3 Initialize $a = (a_1, \ldots, a_n)$ by setting $a_i \leftarrow |x_i - x_{\ell_1}|^z$ for every $i = 1, \ldots, n$;
4 $S \leftarrow \texttt{Initialize}(a)$;
5 **for** $t = 2, \ldots, k$ **do**
6      Choose $r$ uniformly at random from $[0, \texttt{Sum}(S))$;
7      $\ell_t \leftarrow \texttt{Find}(S, r)$;   $a_{\ell_t} \leftarrow 0$;   $i \leftarrow \ell_t - 1$;   $j \leftarrow \ell_t + 1$;
8      **while** $i \geq 0$ **and** $|x_i - x_{\ell_t}|^z < a_i$ **do**
9          $a_i \leftarrow |x_i - x_{\ell_t}|^z$;
10          $i \leftarrow i - 1$;
11      **end**
12      **while** $j \leq n$ **and** $|x_j - x_{\ell_t}|^z < a_j$ **do**
13          $a_j \leftarrow |x_j - x_{\ell_t}|^z$;
14          $j \leftarrow j + 1$;
15      **end**
16      $\texttt{Update}\,(S, a, i + 1, j - 1)$;
17 **end**
18 Sort and re-order $\ell_1, \ldots, \ell_k$ so that $\ell_1 \leq \cdots \leq \ell_k$ ;
19 $i \leftarrow 1$;   $j \leftarrow 1$;
20 **while** $i \leq n$ **do**   /* Assign $x_i$ to the closest center $x_{\ell_{\sigma(i)}}$ among $x_{\ell_1}, \ldots, x_{\ell_k}$. */
21      **if** $j < k$ **and** $|x_i - x_{\ell_j}| \geq |x_i - x_{\ell_{j+1}}|$ **then**
22          $j \leftarrow j + 1$;
23      **else**
24          $\sigma(i) \leftarrow j$;
25          $i \leftarrow i + 1$;
26      **end**
27 **end**
28 **return** $x_{\ell_1}, \ldots, x_{\ell_k}$ and $\sigma$ (converted to the old ordering of $x_1, \ldots, x_n$ before Line 1);

---

The following claim shows that Algorithm 1 correctly implements the $k$-means++ seeding procedure in one dimension.

**Claim C.2.** *Consider the values of $t, a_1, \ldots, a_n$ and the data structure $S$ at the beginning of each iteration of the for-loop (i.e., right before Line 6). Let $s_1, \ldots, s_n$ be the numbers the data structure $S$ keeps track of. For every $i = 1, \ldots, n$, define $p_i := \min_{t'=1,\ldots,t-1} |x_i - x_{\ell_{t'}}|$. Then $s_i = a_i = p_i^z$ for every $i = 1, \ldots, n$. Consequently, the distribution of $\ell_t$ at Line 7 conditioned on the execution history so far satisfies $\Pr[\ell_t = i] = p_i^z / \sum_{i'=1}^n p_{i'}^z$ for every $i = 1, \ldots, n$.*

The claim follows immediately by induction over the iterations of the for-loop based on the description of the data structure $S$ and its operations above. The following lemma bounds the running time of Algorithm 1:

**Lemma C.3** (Lemma 2.4 for general $z \geq 1$). *The expected running time of Algorithm 1 is $O(2^{z/2} n \log n)$. For any $\delta \in (0, 1/2)$, with probability at least $1 - \delta$, Algorithm 1 runs in time $O(2^{z/2} n \log(n/\delta))$. Moreover, Algorithm 1 always runs in time $O(n \log n + nk)$.*

Before proving Lemma C.3, we first use it to prove Theorem C.1.

*Proof of Theorem C.1.* Lemma C.3 bounds the running time of Step 2 of our algorithm PRONE defined in Section 2. Now we show that Step 1 (random one-dimensional projection) can be performed in time $O(\texttt{nnz}(X) + n)$. Indeed, $x_i'$ can be computed as $x_i' = \sum_j x_{ij} v_j$ where the sum is over all the non-zero coordinates $x_{ij}$ of $x_i$, and each $v_j$ is drawn independently from the one-dimensional standard Gaussian (the value of $v_j$ should be shared for all $i$). The time needed to compute $x_1', \ldots, x_n'$ in this way is $O(\texttt{nnz}(X) + n)$. In Step 3 of PRONE, we compute the center of mass for every cluster.

This can be done in time $O(\mathsf{nnz}(X) + n)$ by summing up the points in each cluster and dividing each sum by the number of points in that cluster. $\qquad\square$

The key step towards proving Lemma C.3 is to bound the number of updates to $a$ at Lines 9 and 13. As the algorithm starts by sorting the $n$ input points, which can be done in time $O(n \log n)$, we can assume that the points are sorted such that $x_1 \leq \cdots \leq x_n$. For $i = 1, \ldots, n$ and $t = 2, \ldots, k$, we define $\xi(i, t) = 1$ if $a_i$ is updated at Line 9 in iteration $t$ and define $\xi(i, t) = 0$ otherwise. Here, we denote each iteration of the for-loop beginning at Line 5 by the value of the iterate $t$. We define $u_i := \sum_{t=2}^{k} \xi(i, t)$ to be the number of times $a_i$ gets updated at Line 9. The following lemma gives upper bounds on $u_i$ both in expectation and with high probability:

**Lemma C.4.** *For every $i = 1, \ldots, n$, it holds that*

$$\mathbb{E}[u_i] \leq O(2^{z/2} \log n).$$

*Moreover, for some absolute constant $B > 0$ and for every $\delta \in (0, 1/2)$, it holds that*

$$\Pr[u_i \leq B 2^{z/2} \log(n/\delta)] \geq 1 - \delta.$$

Before proving Lemma C.4, we first use it to prove Lemma C.3.

*Proof of Lemma C.3.* Recall that for $i = 1, \ldots, n$ and $t = 2, \ldots, k$, we define $\xi(i, t) = 1$ if $a_i$ is updated at Line 9 in iteration $t$ and define $\xi(i, t) = 0$ otherwise. Similarly, for $j = 1, \ldots, n$ and $t = 2, \ldots, k$, we define $\xi'(j, t) = 1$ if $a_j$ is updated at Line 13 in iteration $t$ and define $\xi'(j, t) = 0$ otherwise.

The computation at Lines 1-4 takes $O(n \log n)$ time. The computation at Lines 18-28 takes $O(k \log k + n) = O(n \log n)$ time. For $t = 2, \ldots, k$, iteration $t$ of the for-loop takes time $O(\log n + \sum_{i=1}^{n} \xi(i, t) + \sum_{i=1}^{n} \xi'(i, t))$. Summing them up, the total running time of Algorithm 1 is

$$O\left(n \log n + \sum_{i=1}^{n} \sum_{t=2}^{k} \xi(i, t) + \sum_{i=1}^{n} \sum_{t=2}^{k} \xi'(i, t)\right). \tag{9}$$

By Lemma C.4,

$$\mathbb{E}\left[\sum_{i=1}^{n} \sum_{t=2}^{k} \xi(i, t)\right] = \mathbb{E}\left[\sum_{i=1}^{n} u_i\right] = O(2^{z/2} n \log n). \tag{10}$$

Also, for any $\delta' \in (0, 1/2)$, setting $\delta = \delta'/n$ in Lemma C.4, by the union bound we have

$$\Pr\left[\sum_{i=1}^{n} \sum_{t=2}^{k} \xi(i, t) \leq 2B 2^{z/2} n \log(n/\delta')\right] \geq \Pr\left[\sum_{i=1}^{n} \sum_{t=2}^{k} \xi(i, t) \leq B 2^{z/2} n \log(n/\delta)\right]$$
$$\geq 1 - n\delta$$
$$= 1 - \delta'. \tag{11}$$

Similarly to (10) and (11) we have

$$\mathbb{E}\left[\sum_{i=1}^{n} \sum_{t=2}^{k} \xi'(i, t)\right] = O(2^{z/2} n \log n), \quad \text{and} \tag{12}$$

$$\Pr\left[\sum_{i=1}^{n} \sum_{t=2}^{k} \xi'(i, t) \leq 2B 2^{z/2} n \log(n/\delta')\right] \geq 1 - \delta'. \tag{13}$$

Plugging (10) and (11) into (9) proves that the expected running time of Algorithm 1 is $O(2^{z/2} n \log n)$. Choosing $\delta' = \delta/2$ for the $\delta$ in Lemma C.3 and plugging (11) and (13) into (9), we can use the union bound to conclude that with probability at least $1 - \delta$ Algorithm 1 runs in time $O(2^{z/2} n \log(n/\delta))$. Finally, plugging $\xi(i, t) \leq 1$ and $\xi'(i, t) \leq 1$ into (9), we get that Algorithm 1 always runs in time $O(n \log n + nk)$. $\qquad\square$

To prove Lemma C.4, for $i = 0, \ldots, n$ and $u = 0, \ldots, u_i$, we define $t(i, u)$ to be the smallest $t \in \{1, \ldots, k\}$ such that $\sum_{t'=2}^{t} \xi(i, t') = u$. That is, $a_i$ gets updated at Line 9 for the $u$-th time in iteration $t(i, u)$. Our definition implies that $t(i, 0) = 1$ and $t(i, u) \in \{2, \ldots, k\}$ for $u = 1, \ldots, u_i$. We define a nonnegative potential function $\eta(i, u)$ as follows and show that it decreases exponentially in expectation as $u$ increases (Lemma C.5).

**Potential Function** $\eta(i,u)$**.** For $t = 2, \ldots, k$, we consider the value of $\ell_t$ after Line 7 in iteration $t$. For $u = 1, \ldots, u_i$, we define $\eta(i,u)$ to be $\ell_{t(i,u)} - i$, which is guaranteed to be a positive integer by the definition of $t(i,u)$. Indeed, in the while-loop containing Line 9, $i$ starts from $\ell_t - 1$ and keeps decreasing, so whenever Line 9 is executed, $i$ is smaller than $\ell_t$. In particular, $a_i$ is updated at Line 9 in iteration $t(i,u)$ of the for-loop, so we have $i < \ell_{t(i,u)}$. We define $\eta(i,0) = n$, and for $u = u_i + 1, u_i + 2, \ldots$, we define $\eta(i,u) = 0$. See Figure 4 for an example illustrating the definition of $\eta(i,u)$.

$$\ell_{t(i,2)} = i + 1 \qquad \ell_{t(i,1)} = i + 4$$
$$\eta(i,2) = 1 \qquad \eta(i,1) = 4$$

Figure 4: An example illustrating the definition of the potential function $\eta$. Here, $a_i$ is updated at Line 9 for the first time in iteration $t(i,1)$ when $\ell_{t(i,1)} = i + 4$. Then $a_i$ gets updated at Line 9 for the second time in iteration $t(i,2)$ when $\ell_{t(i,2)} = i + 1$. We always have $\ell_t > i$ whenever $a_i$ is updated at Line 9, and $\eta$ is the difference between $\ell_t$ and $i$. Thus in this example $\eta(i,1) = 4$ and $\eta(i,2) = 1$. If $a_i$ is never updated at Line 9 after iteration $t(i,2)$, we define $\eta(i,u) = 0$ for $u = 3, 4, \ldots$.

**Lemma C.5** (Potential function decrease)**.** *For any $i \in \{1, \ldots, n\}$ and $u \in \mathbb{Z}_{\geq 0}$,*

$$\mathbb{E}[\eta(i, u+1)|\eta(i,0), \ldots, \eta(i,u)] \leq \max\left\{0, \frac{2^{z/2}}{2^{z/2}+1}\eta(i,u) - \frac{1}{2}\right\}.$$

Before proving Lemma C.5, we first use it to prove Lemma C.4. Intuitively, Lemma C.5 says that $\eta(i,u)$ decreases exponentially (in expectation) as a function of $u$. Since $\eta(i,u)$ is always a nonnegative integer, we should expect $\eta(i,u)$ to become zero as soon as $u$ exceeds a small threshold. Moreover, our definition ensures $\eta(i, u_i) > 0$, so $u_i$ must be smaller than the threshold. This allows us to show upper bounds for $u_i$ and prove Lemma C.4.

*Proof of Lemma C.4.* Our definition of $\eta$ ensures $\eta(i, u_i) \geq 1$. By Lemma C.5 and Lemma C.7,

$$\mathbb{E}[u_i + 1] \leq \frac{\ln n}{\ln \frac{2^{z/2}+1}{2^{z/2}}} + \frac{1}{1 - \frac{2^{z/2}}{2^{z/2}+1}} = O(2^{z/2}\ln n) + 2^{z/2} + 1.$$

This implies $\mathbb{E}[u_i] = O(2^{z/2}\log n)$. Moreover, by Lemma C.5,

$$\mathbb{E}[\eta(i,u)] \leq \eta(i,0)\left(\frac{2^{z/2}}{2^{z/2}+1}\right)^u = n\left(\frac{2^{z/2}}{2^{z/2}+1}\right)^u,$$

and thus, by Markov's inequality,

$$\Pr[u_i \geq u] = \Pr[\eta(i,u) \geq 1] \leq n\left(\frac{2^{z/2}}{2^{z/2}+1}\right)^u.$$

For any $\delta \in (0, 1/2)$, choosing $u = \ln(n/\delta)/\ln(\frac{2^{z/2}+1}{2^{z/2}}) = O(2^{z/2}\log(n/\delta))$ in the inequality above gives $\Pr[u_i \geq u] \leq \delta$. $\qquad\square$

We need the following helper lemma to prove Lemma C.5.

**Lemma C.6.** *The following holds at the beginning of each iteration of the for-loop in Algorithm 1, i.e., right before Line 6 is executed. Choose an arbitrary $i = 1, \ldots, n$ and define*

$$L := \{i\} \cup \{\ell \in \mathbb{Z} : i < \ell \leq n, |x_i - x_\ell|^z < a_i\}. \tag{14}$$

*Then for $\ell, \ell' \in L$ satisfying $\ell < \ell'$, it holds that $a_{\ell'} \leq 2^z a_\ell$.*

*Proof.* For $t = 2, \ldots, k$, at the beginning of iteration $t$, the values $\ell_1, \ldots, \ell_{t-1}$ have been determined. For every $x \in \mathbb{R}$, let $\rho(x)$ denote the value among $x_{\ell_1}, \ldots, x_{\ell_{t-1}}$ closest to $x$. By Claim C.2, $a_i = |x_i - \rho(x_i)|^z$ for every $i \in [n]$. Now for a fixed $i \in [n]$, define $L$ as in (14) and consider $\ell, \ell' \in L$ satisfying $\ell < \ell'$. It is easy to see that $x_i \leq \rho(x_\ell) \leq x_{\ell'}$ cannot hold because otherwise $a_i \leq |x_i - \rho(x_\ell)|^z \leq |x_i - x_{\ell'}|^z < a_i$, a contradiction. For the same reason, the inequality $x_i \leq \rho(x_{\ell'}) \leq x_{\ell'}$ cannot hold. Thus there are only three possible orderings of $x_i, x_\ell, x_{\ell'}, \rho(x_\ell), \rho(x_{\ell'})$:

1. $\rho(x_\ell) = \rho(x_{\ell'}) < x_i \leq x_\ell \leq x_{\ell'}$;

2. $\rho(x_\ell) < x_i \leq x_\ell \leq x_{\ell'} < \rho(x_{\ell'})$;

3. $x_i \leq x_\ell \leq x_{\ell'} < \rho(x_\ell) = \rho(x_{\ell'})$.

In scenario 3, it is clear that $a_{\ell'} = |x_{\ell'} - \rho(x_{\ell'})|^z \leq |x_\ell - \rho(x_\ell)|^z = a_\ell$. In the first two scenarios, for any $t' = 0, \ldots, t-1$,

$$|x_i - x_{\ell_{t'}}| \geq |x_\ell - x_{\ell_{t'}}| - |x_\ell - x_i| \geq |x_\ell - \rho(x_\ell)| - |x_\ell - x_i| = |x_i - \rho(x_\ell)|.$$

This implies that $\rho(x_\ell)$ is the closest point to $x_i$ among $x_{\ell_1}, \ldots, x_{\ell_{t-1}}$. Therefore, $a_i = |x_i - \rho(x_\ell)|^z$. Jensen's inequality ensures $((g+h)/2)^z \leq (g^z + h^z)/2$ for any $g, h \geq 0$, which implies $(g+h)^z \leq 2^{z-1} g^z + 2^{z-1} h^z$. Therefore,

$$a_{\ell'} \leq |x_{\ell'} - \rho(x_\ell)|^z \leq 2^{z-1}|x_{\ell'} - x_i|^z + 2^{z-1}|x_i - \rho(x_\ell)|^z < 2^z a_i,$$

whereas

$$a_\ell = |x_\ell - \rho(x_\ell)|^z \geq |x_i - \rho(x_\ell)|^z = a_i.$$

Thus, we have $a_{\ell'} \leq 2^z a_\ell$ in all three scenarios. $\qquad\square$

*Proof of Lemma C.5.* Throughout the proof, we fix $i \in \{1, \ldots, n\}$ and $u \in \mathbb{Z}_{\geq 0}$ so that they are deterministic numbers. Algorithm 1 is a randomized algorithm, and when we run it, exactly one of the following four events happens, and we define a random variable $t^*$ accordingly:

1. Event $E_1$: $u_i < u$. That is, $a_i$ gets updated at Line 9 for less than $u$ times. In this case we have $\eta(i, u+1) = 0$ by our definition of $\eta$, and we define $t^* = +\infty$.

2. Event $E_2$: $u_i = u$ and $i$ is never chosen as $\ell_t$ at Line 7. In this case we also have $\eta(i, u+1) = 0$, and we also define $t^* = +\infty$.

3. Event $E_3$: $u_i = u$ and there exists $t \in \{2, 3, \ldots, k\}$ such that $i$ is chosen as $\ell_t$ at Line 7 in iteration $t$. This $t$ must satisfy $t > t(i, u)$, as all updates to $a_i$ in Line 9 must happen before $x_i$ is chosen as a center. We define $t^* = t$ in this case. Again, we have $\eta(i, u+1) = 0$ in this case.

4. Event $E_4$: $u_i > u$. We define $t^* := t(i, u+1) > t(i, u)$ in this case.

Define $E^* := E_3 \cup E_4$. Since $\eta(i, u+1) = 0$ under $E_1$ and $E_2$, it suffices to prove that[5]

$$\mathbb{E}[\eta(i, u+1)|\eta(i, 0), \ldots, \eta(i, u), E^*] \leq \frac{2^{z/2}}{2^{z/2}+1}\eta(i, u) - \frac{1}{2}. \tag{15}$$

By our definition, the random variable $t^*$ takes its value in $\{2, 3, \ldots, k\} \cup \{+\infty\}$. Moreover, $t^* = +\infty$ if and only if $E^*$ does not happen. Therefore, to prove (15), it suffices to prove the following for every $t_0 = 2, 3, \ldots, k$:

$$\mathbb{E}[\eta(i, u+1)|\eta(i, 0), \ldots, \eta(i, u), t^* = t_0] \leq \frac{2^{z/2}}{2^{z/2}+1}\eta(i, u) - \frac{1}{2}. \tag{16}$$

Consider a fixed $t_0 \in \{2, 3, \ldots, k\}$. For $t^* = t_0$ to happen, the following must hold during the execution of Algorithm 1 before iteration $t_0$: $a_i$ has been updated at Line 9 for exactly $u$ times, and $i$ has not been chosen as $\ell_t$ at Line 7. Thus, the rest of the proof assumes that the execution history $H$ of Algorithm 1 before iteration $t_0$ satisfies this property. Now we know that the values $\eta(i, 0), \ldots, \eta(i, u)$ are determined by the execution history $H$. Lines 6-7 guarantee that the distribution of $\ell_{t_0}$ satisfies

$$\Pr[\ell_{t_0} = \ell | H] = \frac{a_\ell}{\sum_{j=1}^n a_j} \quad \text{for every } \ell = 1, \ldots, n,$$

where we use the values $a_1, \ldots, a_n$ right before iteration $t_0$ is executed. Moreover, conditioned on $H$, we have $t^* = t_0$ if and only if $\ell_{t_0} \in L$, where

$$L = \{i\} \cup \{\ell \in \mathbb{Z} : i < \ell \leq n, |x_\ell - x_i|^z < a_i\}.$$

---

[5]We have $\eta(i, u+1) = 0$ also for $E_3$, so one can also simply choose $E^* = E_4$. Choosing $E^* = E_3 \cup E_4$ helps us get improved constants in our bound.

Therefore, if we further condition on $t^* = t_0$, we have $\ell_{t_0} \in L$ and

$$\Pr[\ell_{t_0} = \ell | H, t^* = t_0] = \frac{a_\ell}{\sum_{j \in L} a_j} \quad \text{for every } \ell \in L. \tag{17}$$

When $t^* = t_0$, we have $\eta(i, u+1) = \ell_{t_0} - i$. Therefore, to prove (16), it suffices to show that

$$\mathbb{E}[\ell_{t_0} - i | H, t^* = t_0] \leq \frac{2^{z/2}}{2^{z/2} + 1} \eta(i, u) - \frac{1}{2}. \tag{18}$$

It is clear that we can write $L$ as $L = \{i, i+1, \ldots, \ell^*\}$ for some integer $\ell^* \geq i$. If $u > 0$, Claim C.2 implies $a_i \leq |x_i - x_{\ell_{t(i,u)}}|^z$, and thus $\ell^* < \ell_{t(i,u)}$ and $\ell^* - i \leq \ell_{t(i,u)} - i = \eta(i, u)$. If $u = 0$, we have $\eta(i, u) = n$, so it also holds that $\ell^* - i \leq \eta(i, u)$. By (17) and Lemma C.6, we can set $\gamma = 2^z$ in Lemma C.8 to get

$$\mathbb{E}[\ell_{t_0} - i | H, t^* = t_0] \leq \frac{2^{z/2}}{2^{z/2} + 1}(\ell^* - i) - \frac{1}{2} \leq \frac{2^{z/2}}{2^{z/2} + 1} \eta(i, u) - \frac{1}{2}.$$

This proves (18) and thus proves the lemma. $\qquad \square$

## C.1  Helper Lemmas

**Lemma C.7.** *Let $M \geq 1$ and $\lambda \in (0, 1)$ be parameters. Let $\alpha_0, \alpha_1, \ldots \in [0, +\infty)$ be random variables satisfying $\alpha_0 = M$ and $\mathbb{E}[\alpha_{i+1} | \alpha_1, \ldots, \alpha_i] \leq \lambda \alpha_i$ for every $i = 0, 1, \ldots$. Let $t \geq 0$ be the smallest integer satisfying $\alpha_t < 1$. Then*

$$\mathbb{E}[t] \leq \frac{\ln M}{\ln(1/\lambda)} + \frac{1}{1 - \lambda}.$$

*Proof.* For every $j = 0, 1, \ldots$, we define a random variable $t_j := \min\{t, j\}$. By the monotone convergence theorem, it suffices to show that

$$\mathbb{E}[t_j] \leq \frac{\ln M}{\ln(1/\lambda)} + \frac{1}{1 - \lambda} \quad \text{for every } j = 0, 1, \ldots. \tag{19}$$

We prove (19) by induction on $j$. When $j = 0$, we have $t_j = 0$, and the inequality above holds trivially. We assume that (19) holds for an arbitrary $j \in \mathbb{Z}_{\geq 0}$ and show that it also holds with $j$ replaced by $j + 1$. We have

$$\mathbb{E}[t_{j+1}] = 1 + \mathbb{E}[t_{j+1} - 1] = 1 + \mathbb{E}[\mathbb{E}[(t_{j+1} - 1)|\alpha_1]]. \tag{20}$$

By our definition of $t_{j+1}$, we have $t_{j+1} - 1 = \min\{t - 1, j\}$. Applying our induction hypothesis on the sequence $\alpha_1, \alpha_2, \ldots$, we have

$$\mathbb{E}[(t_{j+1} - 1)|\alpha_1] = \mathbb{E}[\min\{t - 1, j\}|\alpha_1] \leq f(\alpha_1), \tag{21}$$

where

$$f(\alpha) = \begin{cases} \frac{\alpha}{1 - \lambda}, & \text{if } \alpha \in [0, 1); \\ \frac{\ln \alpha}{\ln(1/\lambda)} + \frac{1}{1 - \lambda}, & \text{if } \alpha \geq 1. \end{cases}$$

It is easy to check that $f$ is an increasing concave function of $\alpha \in [0, +\infty)$ and $1 + f(\lambda\alpha) \leq f(\alpha)$ holds for every $\alpha \geq 1$. Plugging (21) into (20), we have

$$\mathbb{E}[t_{j+1}] \leq 1 + \mathbb{E}[f(\alpha_1)] \leq 1 + f(\mathbb{E}[\alpha_1]) \leq 1 + f(\lambda M) \leq f(M) = \frac{\ln M}{\ln(1/\lambda)} + \frac{1}{1 - \lambda}. \quad \square$$

**Lemma C.8.** *Let $\gamma \geq 1$ be a real number. Let $\beta_0, \ldots, \beta_{m-1}$ be non-negative real numbers such that for every $i, j \in \{0, \ldots, m-1\}$ satisfying $i \leq j$, it holds that $\beta_j \leq \gamma \beta_i$. Then,*

$$\sum_{i=0}^{m-1} i\beta_i \leq \left( \frac{\sqrt{\gamma} \cdot m}{\sqrt{\gamma} + 1} - \frac{1}{2} \right) \sum_{i=0}^{m-1} \beta_i.$$

*Proof.* The lemma holds trivially if $\beta_0 = 0$ because in this case $\beta_j \leq \gamma\beta_0 = 0$ for every $j = 0, \ldots, m-1$. We thus assume w.l.o.g. that $\beta_0 > 0$. Define $\tau$ to be the unique real number satisfying

$$\tau \sum_{i=0}^{m-1} \beta_i - \sum_{i=0}^{m-1} i\beta_i = 0.$$

It is clear that $\tau \in [0, m-1]$. Our goal is to prove that

$$\tau \leq \frac{\sqrt{\gamma} \cdot m}{\sqrt{\gamma} + 1} - \frac{1}{2}. \tag{22}$$

Define $\beta_* := \min_{0 \leq i \leq \tau} \beta_i$. For every $i = 0, \ldots, m-1$, we have $\beta_i \geq \beta_*$ if $i \leq \tau$, and $\beta_i \leq \gamma\beta_*$ if $i > \tau$. Therefore, defining $s := \lfloor \tau \rfloor$, we have

$$\begin{aligned}
0 &= \tau \sum_{i=0}^{m-1} \beta_i - \sum_{i=0}^{m-1} i\beta_i \\
&= \sum_{i=0}^{m-1} (\tau - i)\beta_i \\
&\geq \sum_{i \leq \tau} (\tau - i)\beta_* + \sum_{i > \tau} (\tau - i)\gamma\beta_* \tag{23} \\
&= \frac{(s+1)(2\tau - s)}{2} \cdot \beta_* + \frac{(m-s-1)(2\tau - m - s)}{2} \cdot \gamma\beta_*. \tag{24}
\end{aligned}$$

Now, we show that $\beta_* > 0$. For the sake of contradiction, assume $\beta_* = 0$. We already assumed that $\beta_0 > 0$, so $\beta_* \neq \beta_0$. By the definition of $\beta_*$, this means that $\tau > 0$ and inequality (23) is strict, leading to the false claim of

$$0 > \sum_{i \leq \tau} (\tau - i)\beta_* + \sum_{i > \tau} (\tau - i)\gamma\beta_* = 0.$$

Therefore, $\beta_* > 0$ must hold. Now we know that (24) implies

$$(s+1)(2\tau - s) + (m-s-1)(2\tau - m - s)\gamma \leq 0.$$

Treating $s$ as a real-valued variable, the left-hand side is minimized when $s = \tau - 1/2$, giving us

$$(\tau + 1/2)^2 - (m - \tau - 1/2)^2\gamma \leq 0.$$

The inequality above implies

$$(\tau + 1/2)^2 \leq (m - \tau - 1/2)^2\gamma.$$

Taking square root for both sides and solving for $\tau$ gives (22). $\qquad\square$

## D  Data Structure for Fast Sampling in Seeding

In Appendix C, our Algorithm 1 uses a binary tree data structure $S$ that keeps track of $n$ nonnegative numbers $s_1, \ldots, s_n$ and supports several operations. Here, we describe the implementation of this data structure. We assume that $n = 2^q$ for some nonnegative integer $q$. This is without loss of generality because we can choose $n'$ to be the number that satisfy $n \leq n' < 2n$ and $n' = 2^q$ for some $q \in \mathbb{Z}_{\geq 0}$ and consider $s_1, \ldots, s_n, s_{n+1}, \ldots, s_{n'}$ with $s_{n+1} = \cdots = s_{n'} = 0$. Under this assumption, the data structure $S$ is a complete binary tree with $q + 1$ layers indexed by $0, \ldots, q$. In each layer $\zeta = 0, \ldots, q$ there are $2^\zeta$ nodes each corresponding to a set of indices from $\{1, \ldots, n\}$. The root, denoted by $v_1^{(0)}$, is the unique node in layer 0 and it corresponds to the entire set $V_1^{(0)} := \{1, \ldots, n\}$. For $\zeta = 0, \ldots, q-1$, each node $v_j^{(\zeta)}$ in the $\zeta$-th layer has two children $v_{2j-1}^{(\zeta+1)}, v_{2j}^{(\zeta+1)}$ in the $(\zeta+1)$-th layer corresponding to the sets $V_{2j-1}^{(\zeta+1)}, V_{2j}^{(\zeta+1)}$, respectively, where $V_{2j-1}^{(\zeta+1)}$ is the smaller half of $V_j^{(\zeta)}$ and $V_{2j}^{(\zeta+1)}$ is the larger half. Thus,

$$V_j^{(\zeta)} = \{i \in \mathbb{Z} : (j-1)2^{q-\zeta} < i \leq j2^{q-\zeta}\}.$$

Each node $v_j^{(\zeta)}$ in the tree stores a sum $s_j^{(\zeta)} := \sum_{i \in V_j^{(\zeta)}} s_i$.

The data structure supports the four types of operations needed in Appendix C as follows:

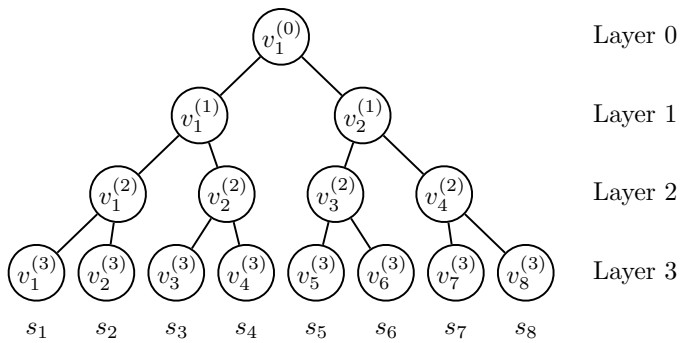

Figure 5: An example of a basic binary tree data structure with $q = 3$.

1. `Initialize`$(a)$. Recursively run `Initialize` on the first half $(a_1, \ldots, a_{n/2})$ and the second half $(a_{n/2+1}, \ldots, a_n)$ to obtain the two subtrees rooted at $v_1^{(1)}$ and $v_2^{(1)}$. Then add a root $v_1^{(0)}$ that stores $s_1^{(0)} \leftarrow s_1^{(1)} + s_2^{(1)}$.

2. `Sum`$(S)$. Simply output $s_1^{(0)}$.

3. `Find`$(S, r)$. If $r < s_1^{(1)}$, recursively call `Find` on the left subtree rooted at $v_1^{(1)}$. Otherwise, recursively call `Find` on the right subtree rooted at $v_2^{(1)}$ with $r$ replaced by $r - s_1^{(1)}$. Once we reach a leaf $v_\ell^{(q)}$, return $\ell$.

4. `Update`$(S, a, i_1, i_2)$. If $i_2 \leq n/2$, recursively call `Update` on the left subtree rooted at $v_1^{(1)}$. If $i_1 > n/2$, recursively call `Update` on the right subtree rooted at $v_2^{(1)}$. Otherwise, we have $i_1 \leq n/2 < i_2$ and we call `Update` on the left subtree with indices $i_1, n/2$ and call `Update` on the right subtree with indices $n/2 + 1, i_2$. In all cases, we update $s_1^{(0)} \leftarrow s_1^{(1)} + s_2^{(1)}$ as the final step. The running time is proportional to the number of nodes we update. We need to update $s_j^{(\zeta)}$ stored at $v_j^{(\zeta)}$ only if $V_j^{(\zeta)} \cap \{i_1, \ldots, i_2\} \neq \emptyset$. For each $\zeta$, the number of such $j$ is at most $(i_2 - i_1 + 1)/2^{q-\zeta} + 2$. Summing up over $\zeta = 0, \ldots, q$, the total number of nodes we need to update is $O((i_2 - i_1 + 1) + q) = O((i_2 - i_1 + 1) + \log n)$.

## E   Additional Experiments and Data

In this section, we provide the tables for speedups and absolute running times of the experiments performed in Section 3. Additionally, we provide some experimental data for the algorithmic approach described in Theorem 2.3.

### E.1   Improved Approximation Ratio

**Experimental Setup.**    This experiment aims to compare the algorithmic approach outlined in Theorem 2.3 to the direct use of `PRONE` as a clustering algorithm as was done in Section 3.2. For this, we use `PRONE` as the approximation algorithm for sensitivity sampling and then cluster the coreset using a weighted variant of the $k$-means++ algorithm. This approach is termed `PRONE` (boosted) in the rest of this section. This pipeline requires as parameters the number of centers $k$ and a hyperparameter $\alpha$ indicating the size of the coreset produced by sensitivity sampling. We aim to compare the clustering cost (see Definition A.1) and running time of our approach to that of $k$-means++.

We run both algorithms on the datasets described in Section 3 and choose $k \in \{10, 25, 50, 100, 250, 500, 1000, 2500, 5000\}$ and $\alpha \in \{0.001, 0.01, 0.1\}$. Each algorithm is run 5 times.

**Results on Improved Approximation Ratio**    *Costs.*   Figure 6 shows the costs of centers produced by this algorithm relative to the cost of centers produced by $k$-means++. It also contains the $(k, 2)$-

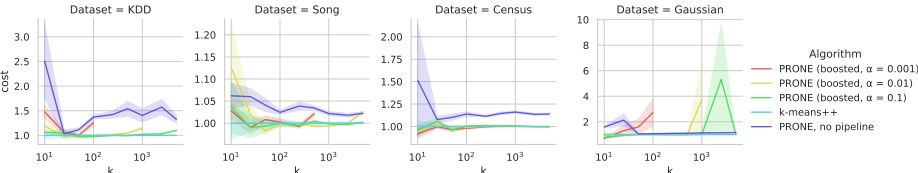

Figure 6: Clustering cost of the boosted variants compared to $k$-means++. Lines in the plot show the cost of centers produced by the boosted algorithm relative to $k$-means++ for centers ranging from 10 to 5000. Dark blue indicates the non-boosted version.

clustering costs of `PRONE` relative to $k$-means++. We can see that on all real datasets, `PRONE` (boosted) produces solutions of the same or better quality than $k$-means++, as long as $\alpha n \ll k$. This shows that although `PRONE` by itself produces centers of worse quality, the `PRONE` (boosted) variant produces centers of the same quality as vanilla $k$-means++. When $\alpha n \approx k$, we observe an uptick in cost before the end of the lines corresponding to $\alpha \in \{0.1, 0.01\}$ in the plots for KDD, Song, and Gaussian. The boosted approach outperforms `PRONE`, which is usually worse by a constant factor compared to the other algorithms, and it helps to reduce significantly the amount of variance in the quality of solutions. On the Gaussian dataset, we observed a failure to sample a point from the central cluster, which explains the spike at $k = 2500$ for the line corresponding to $\alpha = 0.1$.

*Running time.* Table 2 shows the speedup of the boosted approach versus using plain $k$-means++, for the time taken to compute the centers. The running time of our algorithms now scales with $k$, but at a slower rate compared to $k$-means++, as we have to run it on a much smaller dataset. Once again, we observe significant speedups, especially as $k$ grows large.

- As expected, the speedup depends on the choice of the hyperparameter $\alpha$. We observe diminishing returns for larger $\alpha$ as $k$ scales, with the speedup remaining mostly constant for $k \geq 100$ across datasets, except for Gaussian. This is due to the algorithm's running time being dominated by the time it takes to execute $k$-means++ on the coreset, which has $O(ndk)$ asymptotic running time. The speedups we can achieve using this method are significant, up to 118x faster than $k$-means++. We expect that on massive datasets, even greater speedups can be achieved.

- Interestingly, the speedup can come very close to or even out scale $\alpha$, as observed on the KDD and Song datasets. The final stage of the boosted approach executes $k$-means++ on a coreset of size $\alpha n$, so the running time of this step should be $O(\alpha n d k)$. The observed additional speedup may be due to better cache and memory utilization in the $k$-means++ step of the algorithm.

| Dataset | Centers Algorithm | 10 | 25 | 50 | 100 | 250 | 500 | 1000 | 2500 | 5000 |
|---|---|---|---|---|---|---|---|---|---|---|
| Census | PRONE (boosted, $\alpha = 0.001$) | 1.6 | 4.0 | 7.9 | 15.6 | 36.8 | 69.5 | 118.5 | - | - |
| | PRONE (boosted, $\alpha = 0.01$) | 1.5 | 3.7 | 6.7 | 11.7 | 20.8 | 28.5 | 30.1 | 36.2 | 41.9 |
| | PRONE (boosted, $\alpha = 0.1$) | 1.0 | 1.8 | 2.3 | 2.7 | 3.0 | 3.2 | 3.0 | 3.0 | 3.2 |
| | $k$-means++ | 1.0 | 1.0 | 1.0 | 1.0 | 1.0 | 1.0 | 1.0 | 1.0 | 1.0 |
| Song | PRONE (boosted, $\alpha = 0.001$) | 2.1 | 5.3 | 10.7 | 21.3 | 51.7 | 97.5 | - | - | - |
| | PRONE (boosted, $\alpha = 0.01$) | 2.1 | 5.0 | 9.8 | 18.5 | 38.0 | 59.1 | 81.5 | 108.3 | 117.0 |
| | PRONE (boosted, $\alpha = 0.1$) | 1.3 | 2.2 | 2.8 | 3.4 | 3.8 | 4.0 | 4.0 | 4.0 | 3.9 |
| | $k$-means++ | 1.0 | 1.0 | 1.0 | 1.0 | 1.0 | 1.0 | 1.0 | 1.0 | 1.0 |
| KDD | PRONE (boosted, $\alpha = 0.001$) | 1.8 | 5.0 | 9.6 | 20.6 | - | - | - | - | - |
| | PRONE (boosted, $\alpha = 0.01$) | 1.9 | 4.8 | 9.0 | 17.5 | 37.1 | 59.2 | 92.5 | - | - |
| | PRONE (boosted, $\alpha = 0.1$) | 1.4 | 2.7 | 3.6 | 4.9 | 5.7 | 6.2 | 7.1 | 6.6 | 6.4 |
| | $k$-means++ | 1.0 | 1.0 | 1.0 | 1.0 | 1.0 | 1.0 | 1.0 | 1.0 | 1.0 |
| Gaussian | PRONE (boosted, $\alpha = 0.001$) | 0.5 | 0.9 | 1.9 | 3.6 | - | - | - | - | - |
| | PRONE (boosted, $\alpha = 0.01$) | 0.5 | 1.0 | 2.0 | 3.4 | 8.2 | 14.8 | 25.6 | - | - |
| | PRONE (boosted, $\alpha = 0.1$) | 0.4 | 0.8 | 1.4 | 2.2 | 4.0 | 5.0 | 6.1 | 6.8 | 7.2 |
| | $k$-means++ | 1.0 | 1.0 | 1.0 | 1.0 | 1.0 | 1.0 | 1.0 | 1.0 | 1.0 |

Table 2: Average speedup when computing a clustering and assignment for different datasets relative to k-means++. In other words, each cell contains $T_{k\text{-means++}}/T_{\text{PRONE}}$. Missing entries denote the case of $\alpha n > k$.

## E.2 Additional Data

In this section, we provide the running time data for the full range of parameters for the experiments performed in Section 3.2. Table 3 shows the speedups over $k$-means++, analogous to the right-hand-side table in Table 1. Additionally, Table 4 provides absolute running times in milliseconds.

| Dataset | Centers Algorithm | 10 | 25 | 50 | 100 | 250 | 500 | 1000 | 2500 | 5000 |
|---|---|---|---|---|---|---|---|---|---|---|
| Census | PRONE | 1.5 | 3.8 | 7.5 | 15.1 | 36.2 | 73.2 | 142.2 | 351.9 | 662.5 |
| | PRONE (variance) | 0.5 | 1.1 | 2.2 | 4.6 | 11.0 | 22.2 | 43.7 | 109.5 | 214.7 |
| | PRONE (covariance) | 0.2 | 0.5 | 1.1 | 2.2 | 5.2 | 10.7 | 21.0 | 54.7 | 117.4 |
| | $k$-means++ | 1.0 | 1.0 | 1.0 | 1.0 | 1.0 | 1.0 | 1.0 | 1.0 | 1.0 |
| Song | PRONE | 2.0 | 5.0 | 9.7 | 19.1 | 46.1 | 95.5 | 188.2 | 443.0 | 837.5 |
| | PRONE (variance) | 0.5 | 1.1 | 2.3 | 4.5 | 11.4 | 23.1 | 44.2 | 110.9 | 217.2 |
| | PRONE (covariance) | 0.2 | 0.4 | 0.8 | 1.5 | 4.0 | 8.2 | 15.5 | 40.0 | 82.4 |
| | $k$-means++ | 1.0 | 1.0 | 1.0 | 1.0 | 1.0 | 1.0 | 1.0 | 1.0 | 1.0 |
| KDD | PRONE | 1.5 | 3.7 | 6.9 | 16.3 | 39.5 | 68.3 | 158.5 | 414.7 | 727.5 |
| | PRONE (variance) | 0.7 | 1.7 | 3.1 | 6.3 | 16.1 | 32.0 | 63.4 | 159.6 | 312.4 |
| | PRONE (covariance) | 0.3 | 0.7 | 1.3 | 2.6 | 6.8 | 12.9 | 25.8 | 58.5 | 128.4 |
| | $k$-means++ | 1.0 | 1.0 | 1.0 | 1.0 | 1.0 | 1.0 | 1.0 | 1.0 | 1.0 |
| Gaussian | PRONE | 0.5 | 1.0 | 1.9 | 3.8 | 9.2 | 18.3 | 35.7 | 85.3 | 165.9 |
| | PRONE (variance) | 0.5 | 1.0 | 2.0 | 3.8 | 9.1 | 17.7 | 34.5 | 83.6 | 162.9 |
| | PRONE (covariance) | 0.4 | 1.0 | 1.7 | 3.6 | 8.2 | 16.1 | 31.8 | 79.9 | 152.6 |
| | $k$-means++ | 1.0 | 1.0 | 1.0 | 1.0 | 1.0 | 1.0 | 1.0 | 1.0 | 1.0 |

Table 3: Average speedup when computing a clustering and assignment for different datasets relative to k-means++. In other words, each cell contains $T_{k\text{-means++}}/T_{\text{PRONE}}$.

| Dataset | Centers Algorithm | 10 | 25 | 50 | 100 | 250 | 500 | 1000 | 2500 | 5000 |
|---|---|---|---|---|---|---|---|---|---|---|
| Census | PRONE | $525.1 \pm 15.3$ | $534.3 \pm 23.4$ | $534.6 \pm 20.8$ | $530.1 \pm 15.4$ | $538.9 \pm 15.2$ | $533.9 \pm 11.7$ | $547.4 \pm 23.8$ | $548.2 \pm 13.5$ | $563.7 \pm 8.7$ |
| | PRONE (variance) | $1750.6 \pm 71.7$ | $1778.9 \pm 38.1$ | $1780.3 \pm 23.9$ | $1752.7 \pm 56.9$ | $1768.6 \pm 28.2$ | $1757.6 \pm 37.8$ | $1778.9 \pm 27.5$ | $1761.0 \pm 73.3$ | $1739.6 \pm 10.9$ |
| | PRONE (covariance) | $3769.0 \pm 152.2$ | $3882.5 \pm 167.8$ | $3743.9 \pm 316.7$ | $3714.2 \pm 378.3$ | $3766.4 \pm 220.2$ | $3662.9 \pm 122.6$ | $3708.7 \pm 372.4$ | $3525.6 \pm 609.9$ | $3182.2 \pm 288.7$ |
| | $k$-means++ | $812.5 \pm 20.9$ | $2040.6 \pm 25.6$ | $3994.3 \pm 25.3$ | $7992.9 \pm 91.5$ | $19519.0 \pm 161.5$ | $39058.4 \pm 337.8$ | $77823.6 \pm 527.5$ | $192913.1 \pm 3657.0$ | $373488.2 \pm 221.9$ |
| Song | PRONE | $104.4 \pm 5.5$ | $101.8 \pm 5.2$ | $106.5 \pm 4.1$ | $109.9 \pm 6.8$ | $113.7 \pm 8.8$ | $109.0 \pm 6.4$ | $108.7 \pm 3.9$ | $117.2 \pm 6.3$ | $120.8 \pm 12.9$ |
| | PRONE (variance) | $443.3 \pm 5.0$ | $448.3 \pm 8.0$ | $450.1 \pm 9.9$ | $468.0 \pm 31.6$ | $458.8 \pm 15.1$ | $450.7 \pm 10.0$ | $462.6 \pm 11.6$ | $468.3 \pm 14.0$ | $465.7 \pm 14.9$ |
| | PRONE (covariance) | $1164.4 \pm 162.1$ | $1266.5 \pm 136.6$ | $1332.4 \pm 116.3$ | $1381.8 \pm 108.4$ | $1322.2 \pm 173.4$ | $1263.0 \pm 175.6$ | $1324.2 \pm 108.4$ | $1296.8 \pm 177.8$ | $1228.0 \pm 175.2$ |
| | $k$-means++ | $207.4 \pm 4.0$ | $513.7 \pm 9.8$ | $1036.6 \pm 17.0$ | $2103.9 \pm 44.2$ | $5245.9 \pm 143.6$ | $10412.5 \pm 119.3$ | $20464.1 \pm 196.9$ | $51917.4 \pm 554.3$ | $101149.0 \pm 1962.7$ |
| KDD | PRONE | $31.2 \pm 5.2$ | $32.2 \pm 7.4$ | $34.1 \pm 1.8$ | $28.9 \pm 4.9$ | $29.2 \pm 9.1$ | $34.6 \pm 6.3$ | $30.8 \pm 8.6$ | $28.8 \pm 5.2$ | $33.4 \pm 6.6$ |
| | PRONE (variance) | $72.0 \pm 1.4$ | $71.8 \pm 1.4$ | $76.5 \pm 7.2$ | $75.1 \pm 7.1$ | $71.8 \pm 1.2$ | $73.9 \pm 2.1$ | $77.0 \pm 4.6$ | $74.7 \pm 3.2$ | $77.7 \pm 5.9$ |
| | PRONE (covariance) | $169.3 \pm 2.9$ | $173.5 \pm 5.5$ | $180.3 \pm 13.1$ | $184.1 \pm 21.2$ | $170.8 \pm 8.9$ | $182.6 \pm 19.0$ | $188.7 \pm 16.9$ | $204.1 \pm 40.3$ | $189.1 \pm 11.6$ |
| | $k$-means++ | $48.0 \pm 0.3$ | $119.3 \pm 3.7$ | $233.9 \pm 3.2$ | $470.4 \pm 11.6$ | $1153.5 \pm 2.1$ | $2363.3 \pm 108.6$ | $4878.1 \pm 171.1$ | $11928.9 \pm 234.3$ | $24285.9 \pm 243.8$ |
| Gaussian | PRONE | $30.0 \pm 2.3$ | $29.1 \pm 2.2$ | $30.6 \pm 3.3$ | $29.1 \pm 1.4$ | $29.1 \pm 1.0$ | $28.7 \pm 0.2$ | $29.3 \pm 0.4$ | $30.2 \pm 0.6$ | $31.5 \pm 0.6$ |
| | PRONE (variance) | $29.8 \pm 3.0$ | $28.9 \pm 0.3$ | $29.5 \pm 1.0$ | $29.6 \pm 0.6$ | $29.6 \pm 0.3$ | $29.6 \pm 0.4$ | $30.2 \pm 0.3$ | $30.8 \pm 0.2$ | $32.1 \pm 0.6$ |
| | PRONE (covariance) | $31.8 \pm 2.2$ | $31.5 \pm 2.1$ | $34.6 \pm 6.6$ | $30.8 \pm 0.3$ | $32.7 \pm 2.8$ | $32.6 \pm 2.4$ | $32.8 \pm 2.3$ | $32.2 \pm 0.5$ | $34.3 \pm 0.4$ |
| | $k$-means++ | $13.7 \pm 2.2$ | $30.0 \pm 0.4$ | $59.5 \pm 2.2$ | $111.5 \pm 2.8$ | $268.6 \pm 1.5$ | $525.3 \pm 1.9$ | $1043.3 \pm 1.4$ | $2575.3 \pm 5.0$ | $5229.6 \pm 135.0$ |

Table 4: Average running time and standard deviation in milliseconds when computing a clustering and assignment for different datasets relative to k-means++.

