# OpenReview forum: "Simple, Scalable and Effective Clustering via One-Dimensional Projections"
_NeurIPS.cc/2023/Conference — NeurIPS 2023 poster_

### Official Review · Reviewer_fbAv · 2023-06-10

**Soundness:** 3 good
**Presentation:** 3 good
**Contribution:** 2 fair
**Rating:** 5
**Confidence:** 4

**Summary:**

$k$-means clustering is a classical and fundamental method in machine learning. This paper proposes a very simple method that projects the $n$ data points onto a one-dimensional space by generating a random Gaussian vector. Moreover, the one-dimensional projections can be used for coreset construction and the seeding in $k$-means++. The experimental results have proven the efficiency of the proposed algorithm.

**Strengths:**

(1) The proposed method is very simple and easy to understand.
(2) The idea of projecting the $n$ input points onto one-dimensional space is kind of novel.
(3) This paper provides strict proofs for the running time and approximation ratio.
(4) Experimental results show the efficiency for coreset construction and $k$-means++ implementation.

**Weaknesses:**

(1) The essence of the proposed method is dimension reduction. To be specific, the pairwise distances of the $n$ data points are roughly preserved by projecting onto one-dimensional space. The projected distances are used to $k$-means++ seeding and an approximate clustering. By the sorted $n$ scalars, the seeding can be implemented in time $O(n \log{n})$ independent of $k$ but at the cost of $\widetilde{O}(k^4)$ approximation ratio. For me, I think the technique is straightforward and the idea's novelty is a little limited.
(2) The colors of the curves in Figure 2 and Figure 3 are not easy to distinguish.

**Questions:**

The approximation ratio $\widetilde{O}(k^4)$ can be improved to $O(\log{k})$ with running time $O(\text{nnz}(X) + n\log{n}) + \text{poly}(kd) \log{n}$. However, why does not this result show in the experiments?

**Limitations:**

See the Weaknesses part.

---

> ### Author Rebuttal · Authors · 2023-08-10
>
> Thank you for your helpful feedback! Here is our response to the individual comments and questions in your review:
>
> **Dimension reduction techniques.** We would like to emphasize that our dimension reduction analysis is much more sophisticated than simply showing that the pairwise distances between the n data points are roughly preserved. As we discuss at Line 221, it is *not possible* to guarantee that all pairwise distances between the n data points are preserved up to a factor better than poly(n) after a one-dimensional projection. Nevertheless, we prove a much better approximation ratio bound poly(k) for the k-means objective. As we mention at Line 229, our analysis follows the approach of [21] (Cohen et al. ‘15) which involves controlling the pairwise distances between only the k optimal centers rather than the entire dataset. Moreover, in [21] and other previous work on dimension reduction for clustering (see Line 215), the target dimension is at least $\\Omega(\\log k)$. We consider an extremely low target dimension, namely one dimension, in order to facilitate a very fast algorithm after dimension reduction for constructing the final clustering. See Lemma B.3 for details on the precise properties of one-dimensional projections we use. The running time analysis of our algorithm after dimension reduction is also far from being straightforward and it requires a careful potential function argument (see Lemma C.5).
>
> **Experiments on improved approximation ratio.** We achieve the improved approximation ratio by using the initial clustering computed by our fast algorithm to create a coreset, and then running k-means++ on the coreset. Our Section 3.1 is exactly dedicated to experiments that implement this coreset pipeline to improve the approximation ratio. We will make this clearer in the final version.
>
> We also performed additional experiments, in which we ran the version with a reduced approximation ratio and compared it to k-means++, see the rebuttal pdf for a plot which shows the relative cost of the centers produced by this variant for three choices of the coreset size hyperparameter (relative coreset size vs original dataset size). We found that the clustering costs are usually within 10% of full k-means++, while on the KDD dataset it can be as much as 30%. This variant can be slower than k-means++ on smaller datasets for small $k$, but on large datasets such as Census and Song and for $k > 100$ we achieve speedups of 20-100, when choosing the hyperparameter to be 0.01 or lower.
>
> **Colors in the figures.** Thanks for the feedback! We will make sure to adjust the colors in Figures 2 and 3 to make them more distinct in the final version.

---

> > ### Comment · Reviewer_fbAv · 2023-08-14
> >
> > Thank you for your clear response!
> > I have another question: your experiments display coreset constructions and comparisons with $k$-means++; how about comparing the proposed method with some related works, for example, the $k$-means clustering methods based on dimension reduction, with respect to running time and approximation quality?
> >
> > Some minor errors:
> > Line 55, a -> an
> > Line 60, a -> an
> > Line 230, who -> which
> > Line 306, has -> have

---

> > > ### Author Response · Authors · 2023-08-16
> > >
> > > Thank you for replying and pointing out typos!
> > >
> > > Given the extensive prior literature on clustering algorithms, we had to make a judicious choice of which algorithms to compare our new algorithm to. We wanted to explain why we chose to conduct the specific experiments we report in our paper, and why the results of the experiments gave us confidence about the value of our algorithm compared to other algorithms in the literature.
> > >
> > > As explained below, our main takeaway is that our experiments (1) confirm the superior running time of our algorithm, and (2) show that our algorithm can be used to obtain clusters with good quality (in particular, we justify our choice of k-means++ as a strong benchmark).
> > >
> > > **Running time.** Our algorithm is very simple to implement (code in supplementary) and its running time O(nnz(X) + n log n) is nearly linear in the size of the input dataset. That is, its running time is not much larger than the time needed to read the input, even when the input is sparse. As far as we know, no previous clustering algorithm achieves a similar or better running time. While we can use previous dimension reduction algorithms to reduce the data dimension in time as little as O(nnz(X)), running a clustering algorithm, say k-means++, after dimension reduction still requires O(nd’k) additional time, where d’ (=Omega(log k) in prior work) is the reduced dimension. The key point is that our running time does not have the multiplicative factor k, so we expect our algorithm to run much faster, especially for large k.
> > >
> > > **Clustering quality.** In our experiments, we compare the clustering quality of our algorithm with k-means++, which is the algorithm of choice in practice. While there are several dimension reduction approaches for clustering as you point out, each of these still needs a clustering algorithm to be applied after dimension reduction. K-means++ is a natural choice for this final clustering step. However, we don’t expect the initial dimension reduction step to actually improve the clustering quality, hence the vanilla k-means++ algorithm is a stronger benchmark for clustering quality.
> > >
> > > We also show (and experimentally confirm) that our algorithm can be used to build coresets with quality comparable to previous coreset algorithms, and we prove a worst-case bound on the *multiplicative* approximation ratio, without the *additive* error in Lightweight coresets.

---

> > > > ### Comment · Reviewer_fbAv · 2023-08-16
> > > >
> > > > Thank you for your responses! I will retain my score.

---

### Official Review · Reviewer_UQjJ · 2023-07-02

**Soundness:** 4 excellent
**Presentation:** 4 excellent
**Contribution:** 3 good
**Rating:** 7
**Confidence:** 4

**Summary:**

The paper proposes a new clustering algorithm that seeks to have the accuracy of k-means++ while avoiding the time complexity of O(ndk). In particular, the paper focuses on settings where k is large and seeks a O(nd+nlogn) time complexity. This is a practically relevant problem.

The main algorithm proposed in the paper is a simple algorithm based on projecting all data points onto a random one-dimensional subspace and performing the k-means++ seeding procedure on the dimensionality-reduced data. Interestingly, it is shown that this can be done in time O(nd+nlog(n)) and it has a poly(k) approximation guarantee with respect to the optimal k-means clustering. While the poly(k) guarantee is worse than the O(log k) guarantee of k-means++, a log(k) approximation guarantee can be obtained if we use the proposed algorithm to build a coreset of size poly(kd)*log(n), and then use k-means++ on the resulting coreset.

While the techniques in [26] (based on approximate nearest neighbor search) achieve similar goals to those of the present paper (a fast alternative to k-means++ with a complexity that doesn’t depend on k), the algorithm in the present paper is attractive for its surprising simplicity.

**Strengths:**

The proposed algorithm is natural and intuitive, but the theoretical analysis is quite involved.

The fact that a one-dimensional projection can be shown to provide an approximation guarantee (although a poly(k) one) is interesting and nontrivial, since a Johnson-Lindenstrauss type approach typically requires a projection onto log(n) dimensions.

After the one-dimensional projection, a standard implementation of k-means++ seeding does not avoid the O(nk) dependence. As such, the paper proposes a careful data structure to perform k-means seeding on a one-dimensional dataset. The proof that this approach runs in time O(n logn) is technically sophisticated and could be of independent interest. The potential function must be carefully defined, and its use to show that each point is only updated O(log n) times is elegant.

The paper is well written and clear, and the results should be of interest to people working on the theory of clustering algorithms and to people looking for practical approaches for performing k-means clustering on large datasets with large k.


**Weaknesses:**

The empirical results in the paper are generally positive, but perhaps a bit underwhelming, as the proposed approach doesn’t seem to have a strong advantage over the Lightweight coreset approach from [10]. In particular, the proposed algorithm is slower than the Lightweight approach, and the accuracy of the two approaches is similar (except on the Gaussian dataset case, which was designed to adversarially exploit a weakness of Lightweight).

I found it a little confusing to distinguish between the different variants of the algorithm, and it may be helpful to give them clear names (rather than saying “Ours”). In particular, in Section 1, there is a brief discussion about how the approximation can be reduced from poly(k) to O(log k), by incurring an additional poly(nk)*log(n) time. Are the results in 3 and Table 1 for the variant with the O(log k) approximation? Also, in section 1, it says that this improvement is further discussed in the supplementary material, but it wasn’t very clear where. Is this just referring to the general discussion on coresets in Appendix A? It may be useful to have an explicit theorem about the algorithm that combines the proposed algorithm with k-means++ on a coreset (and that algorithm should have a name). I apologize if I missed this somewhere.

It may be useful to discuss some of the literature on fast clustering algorithms beyond k-means. For instance, there are “fast” implementations of PAM for k-medoids (such as FastPAM, Clarans and BanditPAM) and of dbscan (such as dbscan++). It may be the case that all those incur the O(ndk) time that this paper tries to avoid, but a mention of that would be useful to the community (specially as it relates to the claim that there are no other clustering algorithms with O(n*log(n)) complexity).

Some other minor comments/typos:

- line 144: k-measn -> k-means
- line 278: “compared a” -> “compared to a”
- line 279: “entre dataset” -> “entire dataset”
- line 344: “running independent” -> “running time independent”

**Questions:**

A natural way to perform a one-dimensional reduction for clustering would be to take the first principal component of the dataset and then perform k-means++ on it. Would such a projection be expected to perform better than the random Gaussian projection used? I realize that computing the first PC would probably be slow, but it may be worth discussing. This may be connected with spectral clustering approaches as well, and with the “biased” versions of the proposed algorithm.

A natural question that the proposed algorithm raises is whether one can use a constant number of random projections to improve the poly(k) approximation factor (maybe just reducing the polynomial degree). Could you comment on that?

**Limitations:**

The limitations of the proposed approach are discussed in a fair manner.

---

> ### Author Rebuttal · Authors · 2023-08-10
>
> Thank you for the insightful feedback! Here is our response to the individual comments and questions in your review:
>
> **Advantage over Lightweight coresets.** Compared to lightweight coresets, a crucial property of our algorithm is that it has a rigorous guarantee on its worst-case (multiplicative) approximation ratio. The lightweight coreset instead has an *additional additive factor* in its guarantee, which depends on the clustering cost of the mean of the data. In critical applications, our algorithm avoids the catastrophic failure that is possible with lightweight coresets (for instance, on the Gaussian dataset where the approximation ratio can become arbitrarily large as the clusters are moved away from the origin). Consequently, when working with a massive dataset with unknown structure, it is always safer to use our algorithm compared to lightweight coresets. Additionally, on the KDD-cup dataset, the quality of the coresets produced by our sensitivity sampling based approach appears to be better.
>
> **Variant with reduced approximation ratio.** We briefly discussed how to reduce the approximation ratio of our algorithm at Line 106 in our introduction, and indeed, more discussions are deferred to Appendix A. Specifically, in Theorem A.2 we describe how to build a coreset using a clustering where the size of the coreset depends on the approximation ratio of the clustering. To reduce the approximation ratio of the clustering produced by our algorithm, we apply Theorem A.2 to the clustering to construct a coreset and then run k-means++ on the coreset. Thank you for your suggestion on including a separate theorem about this variant of our algorithm. We will make sure to add the theorem to the final version and include a more accurate pointer to Appendix A. In fact, our Section 3.1 is dedicated to experiments where we apply Theorem A.2 to our algorithm to construct coresets and we compare it with other coreset construction algorithms in the literature. Correspondingly, our results shown in Figure 2 and the left half of Table 1 are about coreset constructions. In contrast, in Section 3.2 we directly run our initial clustering algorithm without reducing the approximation ratio via a coreset, and correspondingly Figure 3 and the right half of Table 1 are about direct implementations of our clustering algorithm and its comparison with k-means++.
>
> We also performed additional experiments, in which we ran the version with a reduced approximation ratio and compared it to k-means++. See the rebuttal pdf for a plot which shows the relative cost of the centers produced by this variant for three choices of the coreset size hyperparameter (relative coreset size vs original dataset size). We found that the clustering costs are usually within 10% of full k-means++, while on the KDD dataset it can be as much as 30%. This variant can be slower than k-means++ on smaller datasets for small $k$, but on large datasets such as Census and Song and for $k > 100$ we achieve speedups of 20-120 times, when choosing the hyperparameter to be 0.01 or lower.
>
> **Algorithms beyond k-means.** Thank you for the great suggestion of surveying fast clustering algorithms beyond k-means. In our final version we will make sure to discuss the prior results you mentioned as well as other related works. After reading about them carefully, we did not find any previous result that shows a running time close to O(nd). All algorithms mentioned either have linear dependence on $k$ (taking $k$ as the number of core points in DBSCAN++) or exhibit running time that grows with $n^2$.
>
> **Using the first principal component.**
> This is an interesting question. The biased and covariance variants of our algorithm are both similar in spirit to your suggestion of projecting to the first principal component. In particular, the covariance variant of our algorithm uses the estimated covariance matrix of the data to sample a vector to project onto. This vector generally captures more of the variance in the data and the clustering (and assignment) produced using this vector generally yields an improvement over sampling uniformly (see e.g. bottom row of Figure 3). When the variance along the first principal component is only slightly larger compared to other directions, projecting deterministically along the first principal component may give suboptimal performance, especially when it leads to (deterministically) collapsing several clusters together.
>
> **Using more than one projection.** Thanks for the great question! We also thought of this natural idea of leveraging more than one projection to improve the approximation ratio, but it seems challenging to achieve this while keeping the running time as small as our current algorithm, especially the running time for producing a clustering after the dimension reduction. Even without considering running time, it is unclear to us whether projecting to O(1) dimensions would improve the approximation ratio significantly (say, by more than a constant factor) over projecting to one dimension. It is certainly interesting to explore further and exploit the optimal relationship between the target dimension and the approximation ratio.

---

> > ### Comment · Reviewer_UQjJ · 2023-08-14
> >
> > Thanks for the careful responses. I would still encourage the authors to try to have more explicit names for the different versions of the algorithms, particularly in figures in tables (but this is just my personal preference). I'm updating my score.

---

### Official Review · Reviewer_t2Mh · 2023-07-06

**Soundness:** 2 fair
**Presentation:** 3 good
**Contribution:** 2 fair
**Rating:** 6
**Confidence:** 2

**Summary:**

The submitted paper focuses on the problem of unsupervised clustering and aims at improving the $\Omega(ndk)$ runtime of the ```kmeans++``` algorithm in the clustering of $n$ $d$-dimensional datapoints in $k$ clusters. The proposed algorithm relies on the intuition of running ```kmeans++``` on a one-dimensional projection of the dataset along a random direction $\boldsymbol v$: the authors prove that such strategy leads to a clustering in $O(nd+n\ln n)$ steps with an approximation ratio scaling as $\tilde O(k^4)$ (to be compared with the $O(\ln k)$ of ```kmeans++```). Numerical experiments compare the performance of the algorithm both with ```kmeans++``` and with other strategies for the production of coresets.

**Strengths:**

The proposed algorithm is very simple to implement (reminiscent of sliced optimal transportation strategies, in which a low-dimensional projection is adopted to solve a high-dimensional problem with a subsequent polynomial gain in speed). The approach proposed by the authors is indeed compatible with a remarkable speed-up and, given its price in quality, suitable for adoption in a pipeline. One of the strengths of the paper is that the authors can provide theoretical guarantees on the performance obtained by this simple trick.

**Weaknesses:**

The proposed algorithm pays the price of better performances producing clusterings of lower quality, as the authors themselves point out. This drawback can be mitigated by adopting the algorithm for the construction of coresets within a pipeline (as discussed in the paper). In the case in which the algorithm is adopted as a stand-alone solution, the (possibly deteriorating) effect of structure in the dataset is only partially investigated and addressed.

**Questions:**

* If I understand correctly, the algorithm relies on a single low-dimensional projection along a random vector $\boldsymbol v\in\mathbb R^d$. I was wondering if the algorithm would benefit from the sampling of $m=O(1)$ vectors, or if instead, such strategy would lead to irrelevant benefits.
* It is not very clear in which units the elements of Table 1 are given (from what I understand the table reports speed-up ratios). Please clarify the caption. Also, do the authors have an estimate of the error on the reported values?
* Do the authors have an intuition about the fact that the biased and covariance version of the algorithm suffers a remarkable slowdown with respect to the unbiased version?
* In section 3.2 it is commented that, when working on the Gaussian dataset, the algorithm *did not pick up one of the points close to the origin* so its output is associated with a particularly high cost. Could the author clarify what they mean with this sentence?

Minor typos
* Line 45: *in approximate* in place of *an approximate*.
* Line 244: *measn* in place of *means*.
* What does *with a few approaches in the literature*  refer to in line 147?
* Line 293: *Senstivity* in place of *Sensitivity*.

**Limitations:**

Limitations of the contribution have been addressed, see also the *Weakness* section.

---

> ### Author Rebuttal · Authors · 2023-08-10
>
> Thank you for your comprehensive review! Here are our answers to your questions:
>
> **Sampling more than one vector.** Thanks for the great question! An obvious way to obtain improved performance by sampling more than one projection vector would be independently running our algorithm multiple times and taking the clustering with the smallest cost. We observed experimentally that using this approach, while taking the minimum of 10 runs with uniformly sampled vectors, produces cluster assignments with cost comparable to the algorithms that sample biased vectors for their projection.
>
> Beyond this obvious approach, it seems challenging to further exploit multiple projections in a sophisticated manner while keeping the running time close to our current algorithm. Even without considering running time, it is unclear to us whether projecting to O(1) dimensions would improve the approximation ratio significantly (say, by more than a constant factor) over projecting to one dimension. It is certainly interesting to further explore and exploit the power of multiple projections.
>
> **Clarifications about Table 1.** Thank you for the feedback, we will make the caption clearer. Additionally, we will add a table showing the absolute running times and their standard deviation in the full version. (see the rebuttal pdf for the corresponding table for Table 1)
>
> **Biased and covariance versions.** The observed slowdown appears to be an idiosyncracy of the method for computing the column-wise standard deviation in the scientific computing library used in our implementation – it is not as optimized as it could be. In the case of the covariance algorithm, the slowdown comes from additionally having to compute a Cholesky factorization.
>
> **High cost on the Gaussian dataset.** In this dataset there is a small cluster of points at the origin, which, if no center is placed in this cluster, gives a very large clustering cost. In one of the executions of the algorithm, this point was not sampled, which is a low probability event due to the randomness of our algorithm.

---

### Official Review · Reviewer_sFfD · 2023-07-07

**Soundness:** 3 good
**Presentation:** 3 good
**Contribution:** 3 good
**Rating:** 6
**Confidence:** 3

**Summary:**

The paper presents a new technique for clustering data such that the running time of the method is independent of the number of clusters, and the approximation error is polynomial only in the number of clusters and not the number of data points.

The paper experimentally tests the method and finds that it is a good balance of speed and quality of the result.

**Strengths:**

There are many strengths.


1) The runtime being independent of $k$ is a significant speed up. Further, the constants seem to be better as well. Since the method is not iterative and only makes a single pass over the data
2) The experimental results on corsets are compelling.
3) The presentation of the method is easy to follow.

**Weaknesses:**

The approximation of ratio of $O(k^4)$ is significantly worse than $O(\log k)$. I was hoping this was only theoretical and that we would see better results experimentally, but the experimental results are not clear.

Specifically for Figure 3, I have no idea what the difference between the bottom and the top row is. This is partly due to the fact that the paper never actually writes down formally the problem or the metrics used. From my understanding, we have $n$ data points $x_1, \ldots, x_n$ in high dimensional space. We want to find $k$ clusters $C_1, \ldots, C_k$ where each $C_i$ is a subset of the $n$ data points such that if $c_i$ is the center of the mass of those points, then

$$ \sum_{i=1}^k \sum_{j \in C_i} \|c_i - x_j\|^2 $$

is minimized. This is what I interpret to be the cost. Hence it is very unclear to me what Figure 3 is showing.



**Questions:**

What is the time complexity for the third step?

---

> ### Author Rebuttal · Authors · 2023-08-10
>
> Thank you for the thoughtful review! Here is our response to the individual comments and questions in your review:
>
> **Experimental results.**
> The experiments in the paper are mainly meant to demonstrate that we achieve significant speedups over the baseline algorithms (see Table 1 in the submission), while achieving results of similar quality. While the centers together with the assignment output by our algorithm do not yield clusterings that are competitive with vanilla k-means++, it provides an extremely fast way of seeding coresets. In additional experiments (see Figure 1 in rebuttal PDF) we found that running k-means++ on this coreset can produce a set of centers with clustering cost very close to running k-means++ on the full dataset. Using it in this configuration still lets us achieve speedups over plain k-means++, especially on large datasets, if many centers have to be chosen.
>
> **Two rows in Figure 3.** Thanks for the feedback! We explain the difference between the bottom and top row below. We will add a formal description of our evaluation metrics to the final version.
>
> Our algorithm computes k cluster centers $c_1,\\ldots,c_k$ as well as an assignment $\sigma:[n]\to [k]$ mapping each input point $x_i$ to a center $c_{\\sigma(i)}$. The bottom row of Figure 3 computes the cost $\\sum_{i=1}^n \\|x_i - c_{\\sigma(i)}\\|_2^2$.
>
> Note that $c_{\\sigma(i)}$ may not be the center closest to $x_i$, and computing the closest center to each $x_i$ requires $O(ndk)$ time in total. The top row of Figure 3 computes the cost where every point $x_i$ is assigned to its closest center: $\\sum_{i=1}^n \\min_{j=1}^k \\|x_i - c_j\\|_2^2$.
>
> **Time complexity of the third step.** The third step which updates each cluster center to the center of mass of the cluster takes O(nnz(X)) time. It can be implemented by summing up all the points in each cluster and then dividing the sum by the size of each cluster. This analysis appears in our proof of Theorem C.1 (Line 801).

---

> > ### Comment · Reviewer_sFfD · 2023-08-12
> > **Thank you for the clarification**
> >
> > I will increase my score

---

### Official Review · Reviewer_7JVT · 2023-07-09

**Soundness:** 2 fair
**Presentation:** 2 fair
**Contribution:** 2 fair
**Rating:** 4
**Confidence:** 4

**Summary:**

This paper introduces a randomized clustering algorithm that provably runs in expected time O(nnz(X) + n log n) for arbitrary k. In theoretical analysis, the proposed algorithm achieves approximation ratio O(k^4 ) on any input dataset for the k-means objective. In the experiments, the quality of the clusters found by the proposed algorithm is usually much better than this worst-case bound. They use the algorithm for k-means clustering and coreset construction. This work shows that the approximation ratio achieved after a random one-dimensional projection can be lifted to the original points and that k-means++ seeding can be implemented in expected time O(n log n) in one dimension.

**Strengths:**

This paper has the following contributions:
It designs a simple, practical algorithm for k-means that runs in time roughly O(nd), independent of k, and produces high-quality clusters. In theoretical analysis, they give a randomized clustering algorithm with provable guarantees on its running time and approximation ratio without making any assumptions about the data. In experiments, they run two types of experiments, highlighting various aspects of the algorithm: Coreset Construction Comparison and Direct k-means++ comparison.

**Weaknesses:**

1.Insufficient research on related work.
There are many methods for accelerating clustering. This paper did not conduct comprehensive research. In the most relevant random projection clustering, this work is not the first random projection clustering work, such as paper [1]. It is not cited in this paper. The superiority of this work cannot be verified. Please compare it from the theoretical analysis, experimental results, algorithm complexity, etc.

[1]Liu W, Shen X, Tsang I. Sparse Embedded $ k $-Means Clustering[J]. Advances in neural information processing systems, 2017.

2.The organization and presentation of this paper can be further improved. The second half of the abstract is logically chaotic.

3.The experiment is insufficient to verify the superiority of this work. The experimental results show that the performance of the proposed algorithm is not excellent, such as in Figure 2,3.

**Questions:**

See “Weaknesses”.

---

> ### Author Rebuttal · Authors · 2023-08-10
>
> Thank you for the helpful feedback! Here is our response to the individual comments in your review:
>
> **Research on related work.** In our submission we extensively discussed prior results on fast algorithms for clustering with a focus on results with provable guarantees. We discussed algorithms in three major categories (Lines 55-72) and other related results (Line 83). We also mentioned previous work on dimension reduction at Line 215. In our final version, we will make sure to cite the paper you suggest [Liu-Shen-Tsang,17] and include a refined discussion about dimension reduction methods for k-means clustering.
>
> While dimension reduction is a major component of our algorithm, our goal is making the *entire* clustering algorithm run fast, rather than *just* the dimension reduction step. For example, while the dimension reduction in [Liu-Shen-Tsang,17] runs very fast in $O(\mathsf{nnz}(X))$ time and reduces the dimension to $d’ = O(k/\\varepsilon^2 + 1 / \\varepsilon^2\\delta)$, running a clustering algorithm such as k-means++ after the dimension reduction would require $O(nd’k) = O(nk^2/\\varepsilon^2 + nk /\\varepsilon^2\\delta)$ time. Our algorithm achieves a running time $O(\\mathsf{nnz}(X) + n\\log n)$ independent of k for the entire clustering algorithm (dimension reduction plus k-means++ seeding afterwards).
>
> **Paper presentation.** Thank you for the feedback! Below is a revised version of our abstract based on your feedback. We welcome any additional suggestions on the presentation of our paper.
>
> Abstract: Clustering is a fundamental problem in unsupervised machine learning with many applications in data analysis. Popular clustering algorithms such as Lloyd's algorithm and $k$-means++ can take $\\Omega(ndk)$ time when clustering $n$ points in a $d$-dimensional space (represented by an $n\times d$ matrix $X$) into $k$ clusters. In applications with moderate to large $k$, the multiplicative $k$ factor can become very expensive. We introduce a simple randomized clustering algorithm that provably runs in expected time $O(\\mathsf{nnz}(X) + n\\log n)$ for arbitrary $k$ and achieves an approximation ratio of $\\smash{\\widetilde{O}(k^4)}$ on any input dataset for the $k$-means objective. Here $\\mathsf{nnz}(X)$ is the total number of non-zero entries in the input dataset $X$, which is upper bounded by $nd$ and can be significantly smaller for sparse datasets. We believe that our theoretical analysis is also of independent interest, as we show that the approximation ratio achieved by a $k$-means algorithm is approximately preserved under random one-dimensional projections and that $k$-means++ seeding can be implemented in expected $O(n \\log n)$ time in one dimension. Finally, we show empirically that our clustering algorithm gives a new tradeoff between running time and cluster quality compared to previous state-of-the-art methods for these tasks.
>
>
> **Experiments.** Figures 2 and 3 show that the clustering cost of the centers produced by our algorithm is comparable to that of k-means++ and that it produces strong coresets of similar quality when used for the initial seeding in state of the art coreset constructions. The main improvement of our algorithm is made in the running time, where we observe large speedups (see Table 1).

---

> > ### Comment · Reviewer_7JVT · 2023-08-21
> >
> > Thank you for your responses!

---

### Author Rebuttal · Authors · 2023-08-10

Here is a one-page pdf containing a figure and a table showing additional experimental results. More details can be found in our responses to the reviewers.

---

### Decision · Program_Chairs · 2023-09-21

**Decision:**

Accept (poster)

**Comment:**

This paper advances the state of the art for clustering.
The paper provides a randomised algorithm which runs intimate independent of k, the number of clusters, and in input sparsity time. This should be of interest to communities involved in clustering large scale data